# Structural basis of frizzled 7 activation and allosteric regulation

Julien Bous [1,4] ✉, Julia Kinsolving [1,4], Lukas Grätz [1], Magdalena M. Scharf [1], Jan Hendrik Voss [1], Berkay Selcuk [2,3], Ogün Adebali [2] & Gunnar Schulte [1] ✉

Frizzleds (ten paralogs: FZD$_{1-10}$) belong to the class F of G protein-coupled receptors (GPCRs), which remains poorly understood despite its crucial role in multiple key biological functions including embryonic development, stem cell regulation, and homeostasis in the adult. FZD$_7$, one of the most studied members of the family, is more specifically involved in the migration of mesendoderm cells during the development and renewal of intestinal stem cells in adults. Moreover, FZD$_7$ has been highlighted for its involvement in tumor development predominantly in the gastrointestinal tract. This study reports the structure of inactive FZD$_7$, without any stabilizing mutations, determined by cryo-electron microscopy (cryo-EM) at 1.9 Å resolution. We characterize a fluctuating water pocket in the core of the receptor important for FZD$_7$ dynamics. Molecular dynamics simulations are used to investigate the temporal distribution of those water molecules and their importance for potential conformational changes in FZD$_7$. Moreover, we identify lipids interacting with the receptor core and a conserved cholesterol-binding site, which displays a key role in FZD$_7$ association with a transducer protein, Disheveled (DVL), and initiation of downstream signaling and signalosome formation.

G protein-coupled receptors (GPCRs) represent the main family of signaling membrane proteins in the animal reign. They are thoroughly studied given their critical involvement in multiple key biological processes, diseases, and their potential as drug targets. The most extensively studied GPCR classes, A, B, and C, contain conserved motifs, allosteric sodium binding sites, and cholesterol-binding sites as well as an intricate network of waters that define the dynamic conformational landscape of the receptor. In the case of Frizzleds (ten paralogs: FZD$_{1-10}$) that are class F GPCRs, these features remain to be elucidated[1–6]. FZDs are involved in multiple signaling pathways including heterotrimeric G protein-dependent signaling[7,8], Disheveled-dependent planar cell polarity (PCP)[9], and Wnt/β-catenin signaling pathways[10] leading to variable cellular outputs[10].

FZD$_7$ in particular is involved in mesendodermal stem cell differentiation[11], mesendoderm migration[12], and plays a crucial role in driving the turnover of the adult intestinal epithelium[13]. In this context, it is important for mediating pathogenic effects of *Clostridioides difficile* toxin B (TcdB) and its entry in colonic epithelial cells, thereby relevant for therapeutic perspectives[14]. Furthermore, FZD$_7$ is upregulated in multiple cancers[15] and plays a central role in several aspects of oncogenesis including tumor proliferation, metastasis, maintenance of cancer stem cells, and chemoresistance[16,17]. In colorectal cancer, FZD$_7$ is highly expressed in cell lines with *APC* or *CTNNB1* mutations, and siRNA-mediated knockdown of FZD$_7$ significantly decreases cell viability and invasion activity of HCT-116 cells in vitro[16]. Thus, FZD$_7$ is an attractive drug target for tumor therapy[18]. Nonetheless, a better overall understanding of FZDs represents a necessary step to efficiently develop potential drugs targeting those receptors preferentially with paralog-selectivity to reduce the risk for unwanted side effects.

[1]Section of Receptor Biology & Signaling, Department of Physiology & Pharmacology, Karolinska Institutet, Stockholm, Sweden. [2]Faculty of Engineering and Natural Sciences, Sabanci University, Istanbul, Turkey. [3]Present address: Department of Microbiology, The Ohio State University, Columbus, Ohio, USA. [4]These authors contributed equally: Julien Bous, Julia Kinsolving. ✉e-mail: julien.bous@ki.se; gunnar.schulte@ki.se

A thorough structural understanding of FZDs and underlying mechanisms of receptor activation is directly correlated to the quantity and quality of structural information available for distinct states. Advances in membrane protein purification, cryogenic electron microscopy (cryo-EM), and data processing have notably accelerated the release of GPCR structures in recent years from the first cryo-EM structure of GPCR-G protein complexes published in 2017[19,20] to the current large quantity of active GPCR structures (https://gpcrdb.org/). Nonetheless, there are only a few structures including information about the transmembrane domains (TM)s of FZDs[8,21–23]. The organization of FZDs spanning different conformations and complexes as well as underlying mechanisms of activation, remain to be properly understood.

Here, we apply state-of-the-art cryo-EM to elucidate the apo structure of FZD$_7$ with an overall resolution of 1.9 Å (FSC 0.143). We took advantage of this high-resolution structure to improve the previously published FZD$_7$-Gs protein model[8] and provide a direct comparison of G protein-bound and apo state of FZD$_7$. Moreover, the cryo-EM map quality of the inactive FZD$_7$ is sufficient to identify an internal water pocket comparable to the previously reported water pockets for class A and B GPCRs[24–27]. We further studied water involvement in FZD$_7$ dynamics and activation by molecular dynamics (MD) simulations. In addition, we identified a conserved cholesterol-binding site involving W[4.50] (Ballesteros–Weinstein numbering), which relates to recent

findings reporting on the involvement of cholesterol in FZD-dependent WNT/β-catenin signaling and potential dysregulation in a pathologic context[28,29].

We probed the importance of this cholesterol-binding site in signaling by employing a combination of site-directed mutagenesis and bioluminescence resonance energy transfer (BRET)-based pharmacological assays. In parallel, we used a sophisticated phylogenetic analysis highlighting the family-wide importance and conservation of regions and residues of interest unraveled by the structural analysis, and notably established that the cholesterol-binding site and the water cavity base represent conserved features in the FZD family.

## Results

### The overall organization of FZD$_7$ in its inactive conformation

The sequence of wild type FZD$_7$ with the addition of a Hemagglutinin signal peptide and a double Strep tag was cloned in a pFastBac vector and expressed in Sf9 insect cells. The receptor was purified as described in ref. 14. In brief, Sf9 pellets were solubilized in LMNG, and FZD$_7$ was further purified by a combination of Strep tag and size exclusion chromatography (SEC) (Superdex 200 increase) (Supplementary Fig. 1a). The fractions corresponding to the FZD$_7$ dimer were subjected to cryo-EM analysis (Supplementary Fig. 1b–f, Supplementary Table 1). FZD$_7$ forms an artificial antiparallel dimer with C$_2$ symmetry (Fig. 1a–b).

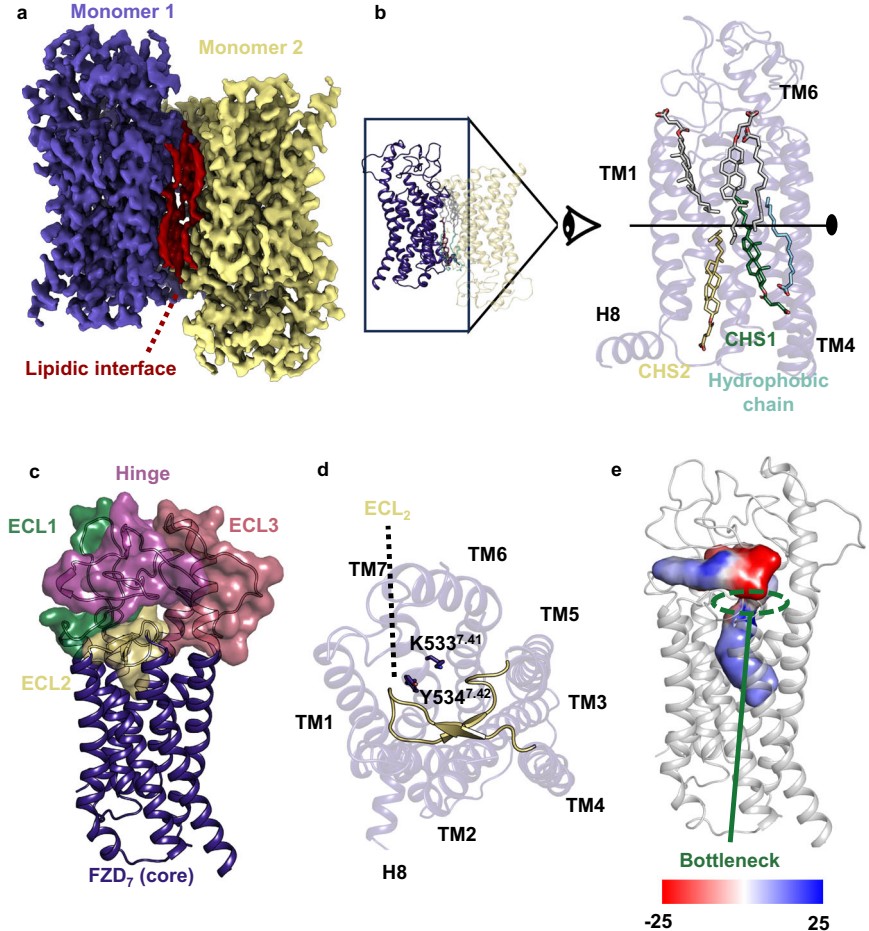

**Fig. 1 | Structure of the inactive FZD$_7$ dimer mediated by a lipidic interface.**
**a** Cryo-EM density map of the FZD$_7$ dimer with C$_2$ symmetry, colored by monomer, with an intermolecular layer of lipids sandwiched between the monomers, shown in red. **b** Atomic arrangement of the FZD$_7$ dimer shown with a close-up side view of the compact interface presenting with endogenous 1-palmitoyl-2-oleoyl-phosphatidylcholine (POPC) and two major cholesterol hemisuccinate (CHS) molecules, which altogether mediate and stabilize the interface between the monomers.

**c** Overall organization of the extracellular region of FZD$_7$ with ECL1, ECL2, ECL3, and the hinge region which altogether form a peripheral lid over the transmembrane bundle. **d** A top view of the helical bundle depicts the folded β-stranded structure of ECL2 with K533[7.41] and Y534[7.42] rendering blockage of the receptor core. **e** Map of the surface electrostatic potential of the internal cavity highlighting the bottleneck in modulating entry of water molecules.

The interface involves TM3-6 and is stabilized by a layer of lipidic aliphatic chains and introduced cholesterol hemisuccinate (CHS), which renders the antiparallel dimer remarkably stable (Fig. 1b, Supplementary Fig. 2). While the density of the aliphatic chains is of poor quality, implying variability and/or a dynamic nature, the high-quality map of CHS1 suggests the presence of a distinct high affinity cholesterol-binding site, while CHS2 displays an intermediate density quality (Supplementary Fig. 2). The CRD of $FZD_7$, part of the linker domain ($M1^{CRD}$-$A205^{linker}$), ICL3 ($T452^{5.74}$-$K463^{6.25}$), and the C terminus ($S565^{8.62}$-$V574^{C-ter}$) were not visible in the density due to flexibility. The $FZD_7$ dimer displays features reminiscent of inactive GPCRs with an inward TM6 position and a densely packed bundle of TMs similar to previously published inactive structures of $FZD_1$, $FZD_3$, $FZD_4$, $FZD_5$ and $FZD_6$ (Fig. 1c Supplementary Fig. 3a)[8,21–23]. The overall position of TM6, however, is closer to TM7 than the structures containing a BRIL in ICL3 (Supplementary Fig. 3b), suggesting that the addition of BRIL fused to TM5 and TM6 by a rigid linker distorts the bottom of TM5-6.

The extracellular loop (ECL2) of $FZD_7$ forms a β-turn partially obstructing the access to the receptor core similarly to other class F receptors (Fig. 1d)[21]. Furthermore, the two bulky residues $K533^{7.41}$ and $Y534^{7.42}$ contribute to obstructing access to the core cavity. Despite these structural elements forming a bottleneck, there is a clear connection between the extracellular side and the core cavity of $FZD_7$ allowing the exchange of water between the two compartments (Fig. 1e). $FZD_7$ includes a hydrophilic internal cavity (volume $1277Å^3$ calculated with CavitOmiX; v. 1.1.beta, 2024, Innophore GmbH) that adopts a bent shape, protruding deep into the receptor core, adapted to the presence of an internal water network important for receptor stability and potentially involved in conformational rearrangements and receptor dynamics (Fig. 1e). ECL1 (E310-E334), ECL3 (V513-P525), and the hinge (F206-Y244) adopt an organized conformation forming the peripheral lid (Fig. 1c). The overall organization of this peripheral lid is variable between the different class F GPCRs (Supplementary Fig. 3a). $FZD_{1,3,6}$ and $FZD_7$ adopt a similar overall organization with (i) a long extension of TM6 above the lipid bilayer like SMO and (ii) a shorter ECL2 that points downward, compared to SMO, where the loop points upward and interacts with the SMO CRD potentially impacting the allosteric cooperation during SMO activation[21,30]. The downward orientation of the $FZD_7$ ECL3 is dictated by a disulfide bridge ($C508^{6.70}$-$C515^{ECL3}$) between the top of TM6 and the ECL3, similar to $FZD_{1,3,6}$ and consistent with the previous hypothesis that the distribution of cysteine residues mediates the cap organization with potential impact on receptor cell surface expression, signaling profiles and receptor specificity[21,31] (Fig. 1c). In comparison, $FZD_4$ features a shorter extension of the TM6 helix, and a shorter hinge that adopts a different peripheral lid organization and $FZD_5$ adopts an intermediate conformation with a slightly shorter TM6 extension compared to $FZD_{1,3,6,7}$ and an intermediate hinge size (Supplementary Fig. 3a).

## $FZD_7$-Gs protein coupling elicits limited conformational rearrangements

A direct comparison of inactive $FZD_7$ and the amended $FZD_7$-Gs protein structure[8] allowed us to investigate the conformational rearrangements involved in constitutive $FZD_7$ activation (G protein coupling) with an unprecedented level of detail generally confirming the previously proposed FZD-G protein coupling mechanism[7,23,32]. Additionally, we elaborated on a hypothesis suggesting that the limited dynamics of TM6 in FZDs explain the relatively poor G protein coupling capacity of FZDs compared to other GPCR classes.

The unliganded $FZD_7$-Gs protein complex presents with a limited outward motion of TM6, a slight inward motion of TM1,2,5 and TM7/H8 upon G protein α5 helix coupling to the intracellular core of the receptor (Fig. 2d, Supplementary Movie 1). The extracellular side of $FZD_7$ maintains the same overall organization in the absence of an agonist irrespective of the receptor's activation status. A kink of the

conserved $P481^{6.43}$ occurs in TM6 (Fig. 2a) along with a rearrangement of a set of residues previously described as the molecular switch $R^{6.32}$-$W^{7.55}$ (Fig. 2f)[7] and the extended molecular switch[32] $W354^{3.43}$–$Y478^{6.40}$ (Fig. 2e) interact by π-π stacking with $W354^{3.43}$ pointing towards TM5 in the inactive state and towards TM7 in the active state (Fig. 2e). The $FZD_7$ $W354^{3.43}$ rotamer flip upon G protein coupling is not observed in the $FZD_{1,3,9}$ structures[23]. The molecular switch $R470^{6.32}$-$W547^{7.55}$ is characterized by the combination of a cation-π interaction and a hydrogen bond (between $R470^{6.32}$ guanidine group and $W547^{7.55}$ backbone carbonyl oxygen) tightly regulating TM6 opening. In this context, G protein coupling promotes TM6 opening with a rotamer flip of $W547^{7.55}$ without fully disrupting the interaction but rather extending it (Fig. 2f). This reorganization is accompanied by a rotamer change of nearby $F474^{6.36}$ (Fig. 2b).

Previously reported class F GPCRs structures display variability in terms of $R/K^{6.32}$-$W^{7.55}$ side chain positioning in both active and inactive states resulting in great variability in $R/K^{6.32}$-$W^{7.55}$ cation-π interaction, ranging from strong interactions to very weak or negligible interactions (Supplementary Table 2). Nonetheless in all structures reported so far $R/K^{6.32}$ can accommodate either cation-π interactions with $W^{7.55}$ or hydrogen bonds with $W^{7.55}$ or the carbonyl oxygen of $T^{7.54}$, $W^{7.55}$, $W^{7.57}$ (Supplementary Table 2), supporting that the molecular switch can undergo drastic conformational changes while preserving TM6/TM7 contacts, thereby regulating and limiting TM6 opening.

For $FZD_7$, to investigate further the dynamics of these two sets of residues ($W354^{3.43}$-$Y478^{6.40}$ and $R470^{6.32}$-$W547^{7.55}$) and the possibility of transient rotamer switches as the main switch mechanism driving FZD activation, we subjected $FZD_7$ in its active and inactive conformation derived from the experimental structures to MD simulations for 300 ns with 3 replicates. Throughout the entire simulations, the respective active and inactive states remain consistent, as evidenced by the TM2-TM6 distances and the TM6 kink observed across all replicates. (Supplementary Figs. 4a–d, 5a, Supplementary Table 3). The dihedral angles of the rotamers $W354^{3.43}$ and $W547^{7.55}$ show a distinct monodispersed profile in the active versus inactive conformation (Fig. 2h, i, Supplementary Figs. 5b, c), suggesting a permanent switch of the two upon heterotrimeric G protein coupling.

G protein coupling also induces rearrangement of ICL1,3, whereas ICL2 maintains the same organization (Fig. 2c, g). ICL1 ($D278^{1.57}$ $S283^{12.51}$) adopts a loose conformation in the inactive $FZD_7$ structure and rearranges into a more compact conformation in the G protein-coupled complex (Fig. 2g). TM5,6 are extended from $R451^{5.73}$ to $D457^{ICL3}$ and $K466^{6.28}$ to $E462^{6.24}$ in the G protein bound $FZD_7$ state, allowing $D457^{ICL3}$, $T459^{ICL3}$ and, $K463^{ICL3}$ to make polar contacts with $G_s$ $α_5$ helix (Fig. 2c).

The mechanisms of TM6 opening are finely regulated among GPCRs with different mechanisms for the different classes. For example, in class A GPCRs a set of shared motifs throughout the transmembrane section rearrange upon agonist stimulation notably inducing a kink of $P^{6.50}$ (25° for the $β_2$ adrenergic receptor ($β_2$AR) PDB:3SN6 (active) PDB:6PS3 (inactive)) provoking a disruption of the ionic lock (consensus motif involving the TM3 D/ERY motif (comprising $D/E^{3.49}$, $R^{3.50}$, and $Y^{3.51}$) and TM2 (for example $T68^{2.39}$ for $β_2$AR) or TM6 ($A^{6.34}$, $E^{6.30}$), involved in controlling the opening of the intracellular segment of TM6 (12 Å for $β_2$AR) (Fig. 3a). The ionic lock is permanently disrupted upon receptor activation[33]. In class B1 GPCRs, sequential peptide and G protein binding induces α helix disruption in TM6 and the formation of a sharp kink (70° for the Glucagon receptor (GCGR) PDB:5XEZ (inactive) PDB:6WPW (active)) with a large outward motion (18 Å for GCGR), leading to a breakage of the conserved polar network $R^{2.46b}$, $R/K^{6.37b}$, $N^{7.61b}$ and $Q^{8.41b}$ (Fig. 3b)[19]. Similarly, other GPCR classes feature their own structural mechanisms regulating TM6 opening[34–36]. Interestingly, the difference in TM6 dynamics between class A ($β_2$AR) (Fig. 3a) and class B1 GCGR (Fig. 3b) GPCRs is correlated to their respective efficacy to couple to and activate the heterotrimeric G

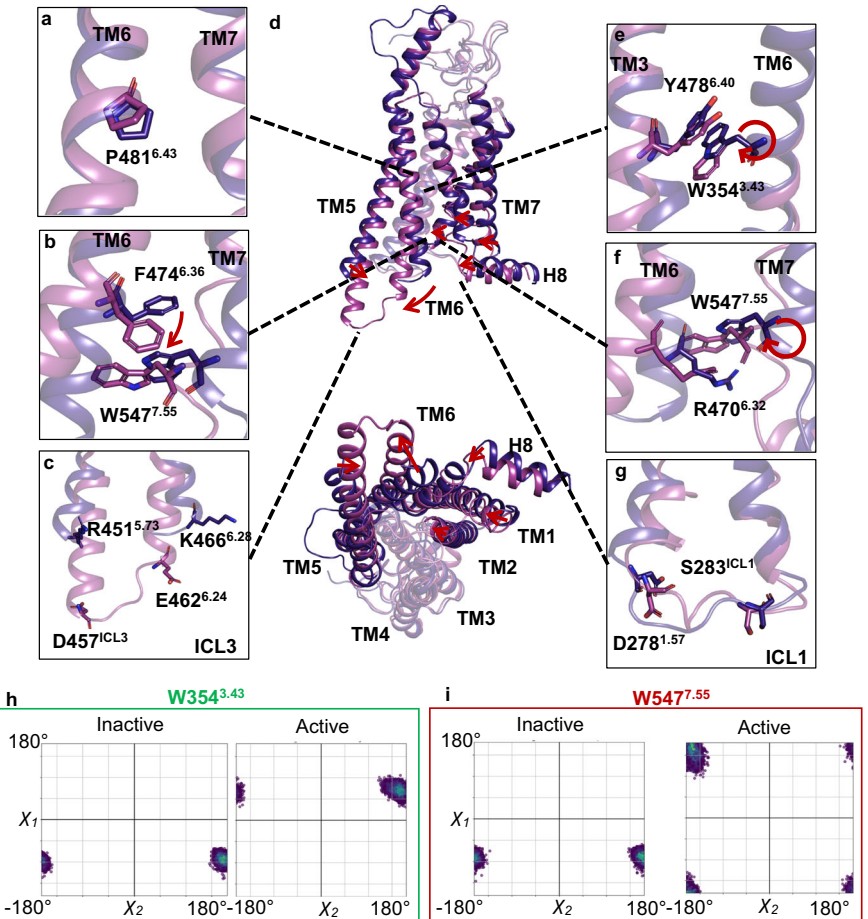

**Fig. 2 | Structural comparison of conserved motifs in the inactive and active FZD₇ conformations.** The intracellular rearrangements of conserved residue **a** P481⁶·⁴³, **b** extended molecular switch residues, F474⁶·³⁶ - W547⁶·³⁶, **e** Y478⁶·⁴⁰ and **f** molecular switch residues W547⁷·⁵⁵ - R470⁶·³² are observed upon agonist binding. **c** A network of extended polar contacts in ICL3 is made between the two conformations of FZD₇. **d** Comparison of the nucleotide-free Gαₛ-bound FZD₇ (pink,

PDB: 7EVW) and inactive FZD₇ (purple) highlighting structural changes of conserved residues and regions of interest. **g** G protein binding rearranges residues in ICL1 involving D278¹·⁵⁷, F282, R281, R280, and M279. Scatterplot of occurring χ1 χ2 dihedral angles of residue **h** W354³·⁴³ and **i** W547⁷·⁵⁵ calculated for each frame (50 ps time steps) over the trajectory from replica 1 of FZD₇ simulations of active and inactive state.

protein with GCGR exhibiting a substantially lower guanine nucleotide exchange activity[37]. This gap in efficacy is due to the higher energy barrier between the active and inactive state for GCGR.

In the case of FZDs, the molecular switch R⁶·³²-W⁷·⁵⁵ acts as a hinge limiter (Fig. 3c, d) permitting only a restricted TM6 opening (11°, 5.5 Å). Hence, the open conformation of FZDs remains suboptimal for G protein coupling providing a rational explanation for the limited capacity of FZDs to couple to heterotrimeric G proteins and their propensity to display selectivity towards DVL over heterotrimeric G proteins[5]. Interestingly, R/K⁶·³² is frequently mutated in cancers with obvious consequences on TM6 dynamics, eventually impacting receptor signaling profiles. FZD₆, R⁶·³²A, R⁶·³²Q, and W⁷·⁵⁵L lose the ability to recruit DVL efficiently confirming the potential to favor a specific signaling pathway in a conformational-dependent manner in cancers[7].

### The structure of FZD₇ contains an internal water pocket that does not rearrange into a channel upon G protein coupling

The internal cavity of FZD₇ can be divided into two sections separated by a bottleneck (Fig. 1e). The bottleneck is formed by a set of residues D405^ECL1, L415^ECL1, Y489⁶·⁵¹, K533⁷·⁴¹, Y534⁷·⁴² (Fig. 4a, b) leaving a sufficient opening to allow water exchange (minimum diameter 4 Å).

The different FZD structures display variable internal pockets (Supplementary Fig. 3c). For example, the structures of inactive FZD₄,₅

contain continuous cavities protruding into the receptor core to a similar depth compared to FZD₇. FZD₄ features a straight pocket due to the less extended peripheral lid[22] and the FZD₅ pocket adopts the same overall bent shape as the one in FZD₇.

The extracellular sections of FZD₁,₃,₆ display a denser packing with a disruption of the cavity preventing potential water exchange between the internal cavity and the extracellular environment. In the case of FZD₁ this is mainly due to the different positioning of the internal side chain network, notably obstructing the bottleneck section. Direct contact between the internal cavity and the extracellular side might still open up upon side-chain reorganization in a physiological context. In the case of FZD₃,₆, the side chains of the Y362^ECL2 (FZD₆) and Y366^ECL2 (FZD₃) block the pocket entrance. In FZD₇, however, the corresponding but less bulky S407^ECL2 (FZD₇) does not obstruct the internal cavity. Furthermore, because of a different hinge folding compared to the one of FZD₇, H181^hinge (FZD₃) and Y173^hinge (FZD₆) also participate in sealing the cavity entrance in FZD₃,₆.

The high-quality data of our FZD₇ structure allowed us to model and characterize an extensive, internal water network in the FZD₇ core (Fig. 4a, Supplementary Fig. 6a, b), similar to the one that plays a crucial role in the folding and dynamics of class A GPCRs[24,25]. The water network in FZD₇ integrally fills the internal cavity, making a large number of polar contacts with inward-facing residues, (Fig. 4a,

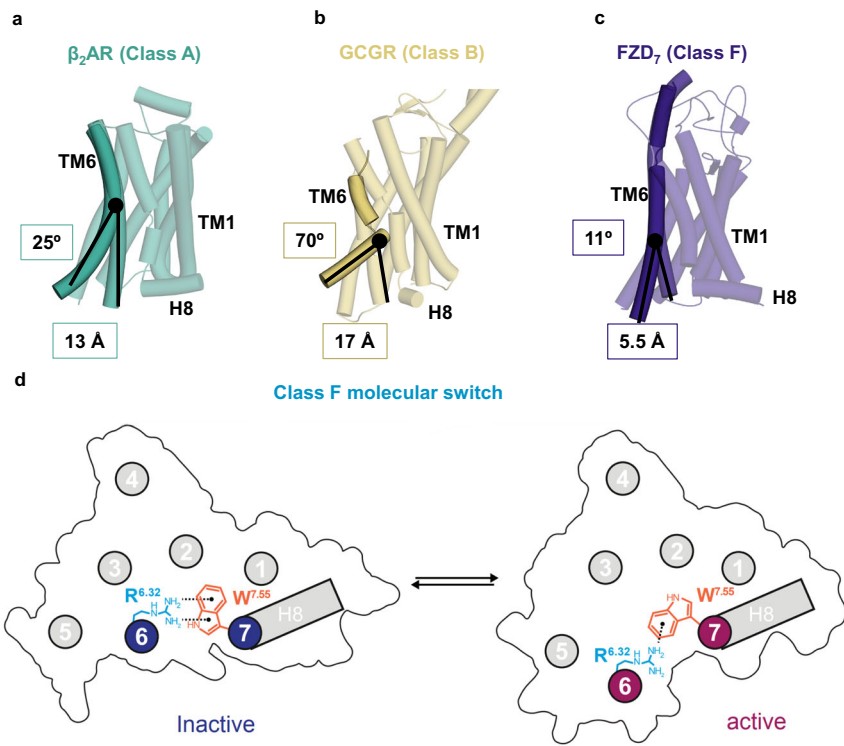

**Fig. 3 | Structural insight into the activation mechanisms across GPCR classes.**
The dynamic nature of TM6 in GPCRs between the inactive and active conformations are shown in **a** class A, β2AR (PDB: 6PS3, 3SN6), **b** class B, GCGR (PDB: 5XEZ and 6WPW) and **c** class F, FZD7 (PDB: 9EPO, 7EVW). **d** The molecular switch in class F, $R^{6.32}$-$W^{7.55}$ acts as a hinge limiter restricting the swing out of TM6.

Supplementary Fig. 6a, b) highlighting its importance in overall structure integrity.

To investigate whether this internal water network is indeed relevant to receptor activation as a conserved, family-wide mechanism, we used an approach combining phylogenetic and structural analyzes. We identified functionally equivalent orthologs (214 on average for each subfamily, 215 for FZD7) for all human class F receptors to determine residues with high class-wide conservation (see Methods). To understand the involvement of conserved residues in receptor activation, we utilized the Residue-Residue Contact Score (RRCS) algorithm (see Methods)[25]. Changes were identified in the contact score (ΔRRCS) in the transition from inactive to G protein-coupled, active FZD7 focusing on residue pairs that define G protein coupling-mediated reorganization. Lastly, we highlighted a subsection of the activation network that is spatially close to water molecules within the provided receptor structure, and identified $C^{1.43}$, $Y^{2.51}$, $W^{3.43}$, $V^{3.44}$, $G^{5.58}$, $Y^{6.40}$, $P^{6.43}$, $M^{7.44}$ and $V^{7.48}$ at the base of the water pocket (Fig. 4c). Additionally, this network consisting of water-interacting residues is immediately connected to the class-wide conserved switch residues $W^{7.55}$ and $F^{6.36}$ that are crucial for receptor activation as we mentioned previously. With validation from the experimental FZD7 structures, $C^{1.43}$, $Y^{2.51}$, $W^{3.43}$, $V^{3.44}$, $G^{5.58}$, $Y^{6.40}$ and $V^{7.48}$ form a solid base in both G protein-coupled, active and inactive FZD structures and prevent drastic reorganization of internal water networks upon G protein coupling, unlike what is observed in class A GPCRs[24,25]. This solid base notably includes $W354^{3.43}$ and $Y478^{6.40}$ presenting a part of the extended molecular switch involved in FZD activation[32]. Due to its tight packing this layer represents a structural hub between TM1-3,5-7 partially mediating the overall bundle organization with the potential to tolerate limited motion of the transmembrane domains while maintaining a similar organization of the extracellular side of FZD7 (Fig. 4d–f; Supplementary Movie 2).

It is also noteworthy that the structural phylogenetic analysis indicates SMO lacks the conserved molecular mechanism found in FZDs (Fig. 4c). The difference in conservation of residues between FZDs and SMO in the receptor core creates a basis for the different activation mechanism observed between FZDs and SMO despite their evolutionary relatedness.

## Dynamics of the internal water pocket in FZD7

To further characterize the role of the internal water network in G protein coupling, we analyzed the temporal distribution of waters in the internal pocket of FZD7 in both the active and inactive conformation by MD simulations. Most water positions attributed to the inactive structure are present in the simulation with variable occupancy (Fig. 5a). The water density maps with an occupancy threshold of 20% show that the entire pocket is occupied and continuously filled with water (Fig. 5b,c), without significant depth differences between active and inactive conformations in agreement with the respective internal pockets of the experimental structures. The TM6 outward motion allows intracellular water to occupy a shallow groove in the cavity normally occupied by the α5 helix of the heterotrimeric G protein, but the solid water pocket base remains tight preventing water exchange with the upper cavity (Fig. 5b, c). To investigate the dynamics of the water pocket further, we examined how minor rearrangements in the side chain orientation could further restrict the bottleneck, as seen in the FZD1 apo structure[23]. We investigated the distances between atoms of key residues of the bottleneck ($L415^{ECL2}$-$K533^{7.41}$ and $D405^{ECL2}$-$L415^{ECL2}$) to better understand this process and to probe the dynamics of the bottleneck (Fig. 5f, Supplementary Figs. 7a–c). Independent of the activation state, the bottleneck adopts two major sub-conformations (Fig. 5d–f). In the first conformation, the bottleneck forms a tight ring transiently disrupting the cavity and opening with slight side chain conformational changes (Fig. 5d). The second

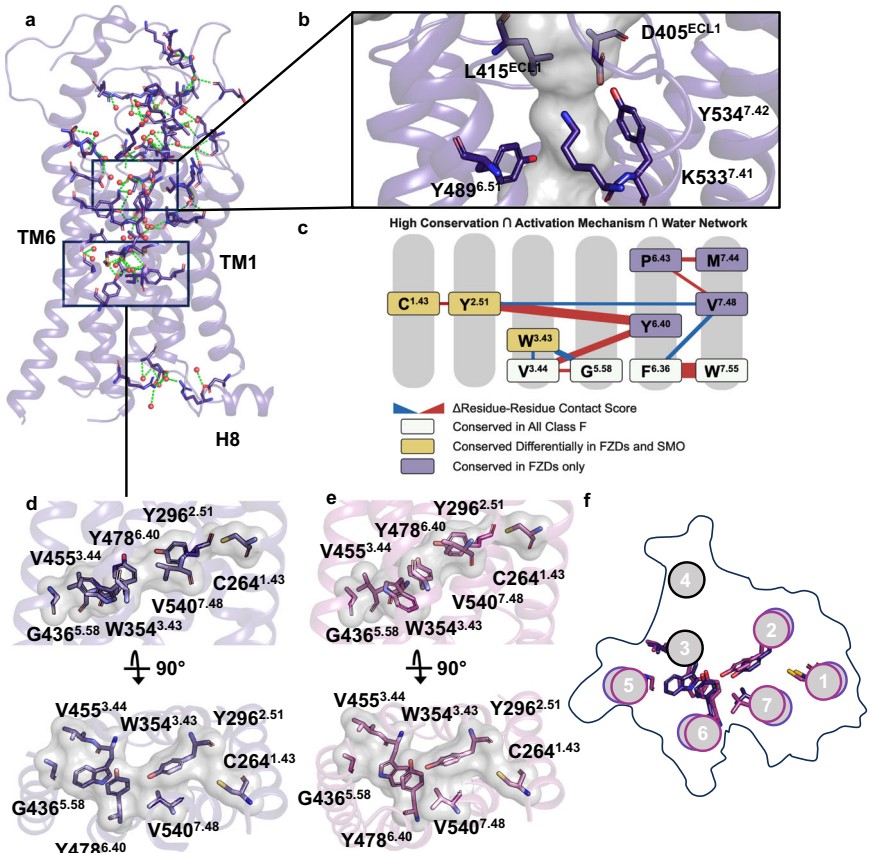

**Fig. 4 | Water-mediated interactions and conserved residues of FZD₇. a** Overall organization of the internal water pocket of inactive FZD₇. **b** The region of the cavity referred to as the bottleneck is comprised of the following residues: L415^ECL1, D405^ECL1, Y489^6.51, Y534^7.42, and K533^7.41. **c** Phylogenetic analysis of FZD₇ depicting highly conserved and interacting residue pairs involved in the internal water pocket. **d, e, f** The tight base of the internal water cavity facilitates a reorientation of residues W354^3.43, Y478^6.40, Y296^2.51, and V540^7.48 in the **d** inactive conformation and **e** active conformation. **f** A top view of the transmembrane domains between the two structures highlighting residues of interest.

conformational state shows a reorganization of the ECL2 with side chain reorientation of D405^ECL2 and L415^ECL2, causing a pocket disruption upward towards the bottleneck position (Fig. 5e). These two states are transient and were observed in both active G protein-bound and inactive state-derived simulations (Supplementary Figs. 7b,c; active replicate 3 and inactive replicate 1). While these variations of the extracellular section of the receptor are not directly related to G protein coupling, they might play an important role in other signaling events, such as FZD-DVL coupling or agonist stimulation, which remain structurally characterized.

## A conserved and functionally important cholesterol-binding site in FZDs

While the structural analysis revealed a well-defined cholesterol-binding site in FZD₇, we also took a structure-independent computational approach that only utilizes the evolutionary information to determine if a cholesterol-binding pocket is conserved in this family (Fig. 6a). Similar to the water network analysis, we calculated residue conservation for all class F receptors (Fig. 6a, Supplementary Fig. 8a, b). We identified residues that are conserved within paralogs with no sequence variation or variation with only similar amino acids (BLOSUM80 score greater than 2). We hypothesized that a set of conserved and structurally continuous amino acids on the receptor surface is associated with cholesterol binding. We could only identify a single region strongly enriched in conserved aromatic residues (Fig. 6a). Upon projection of those residues onto a receptor structure (Fig. 6a), we observed that residues at the receptor surface were exclusively

enriched at the structurally identified cholesterol-binding-site. This exclusive enrichment highlights the importance of cholesterol binding at this conserved site not only for FZD₇ but for all class F GPCRs. Our analysis revealed five aromatic residues constituting a binding surface for cholesterol. While residues W^4.50, H^4.46 and F^3.35 are fully conserved, the positions F/Y^2.46 and F/Y^3.34 are conserved in their aromatic nature in all class F receptors. Thus, the results from the class-wide phylogenetic analysis are in agreement with the structural analysis pinpointing five conserved, aromatic residues at the receptor surface between TM2-4 that coincide with the cholesterol-binding site observed in the FZD₇ structure.

Cholesterol-binding motifs in membrane proteins have been extensively studied[38]. Notably, the cholesterol recognition/interaction amino acid consensus sequence (CRAC) domain[39], and cholesterol consensus motifs (CCMs) are present in class A GPCRs[40]. The class A CCMs and the highly conserved cholesterol-binding site found in FZDs share the same position when projected on the overall topology of the receptor (between TM3 and TM4). Additionally, W^4.50, which participates in cholesterol-π stacking within the class A CCMs, is also conserved in class F GPCRs. Unlike conventional class A CCMs, the FZD cholesterol-binding site is heavily enriched in aromatic residues emphasizing a strong potential for cholesterol-π stacking compensation mechanism, suggesting a peculiar functional importance for cholesterol in FZDs. While cholesterol is important for folding and stabilization in class A GPCRs, we hypothesize that cholesterol-mediated receptor stabilization presents a conserved feature of FZDs and that cholesterol plays a role in receptor activation and signaling.

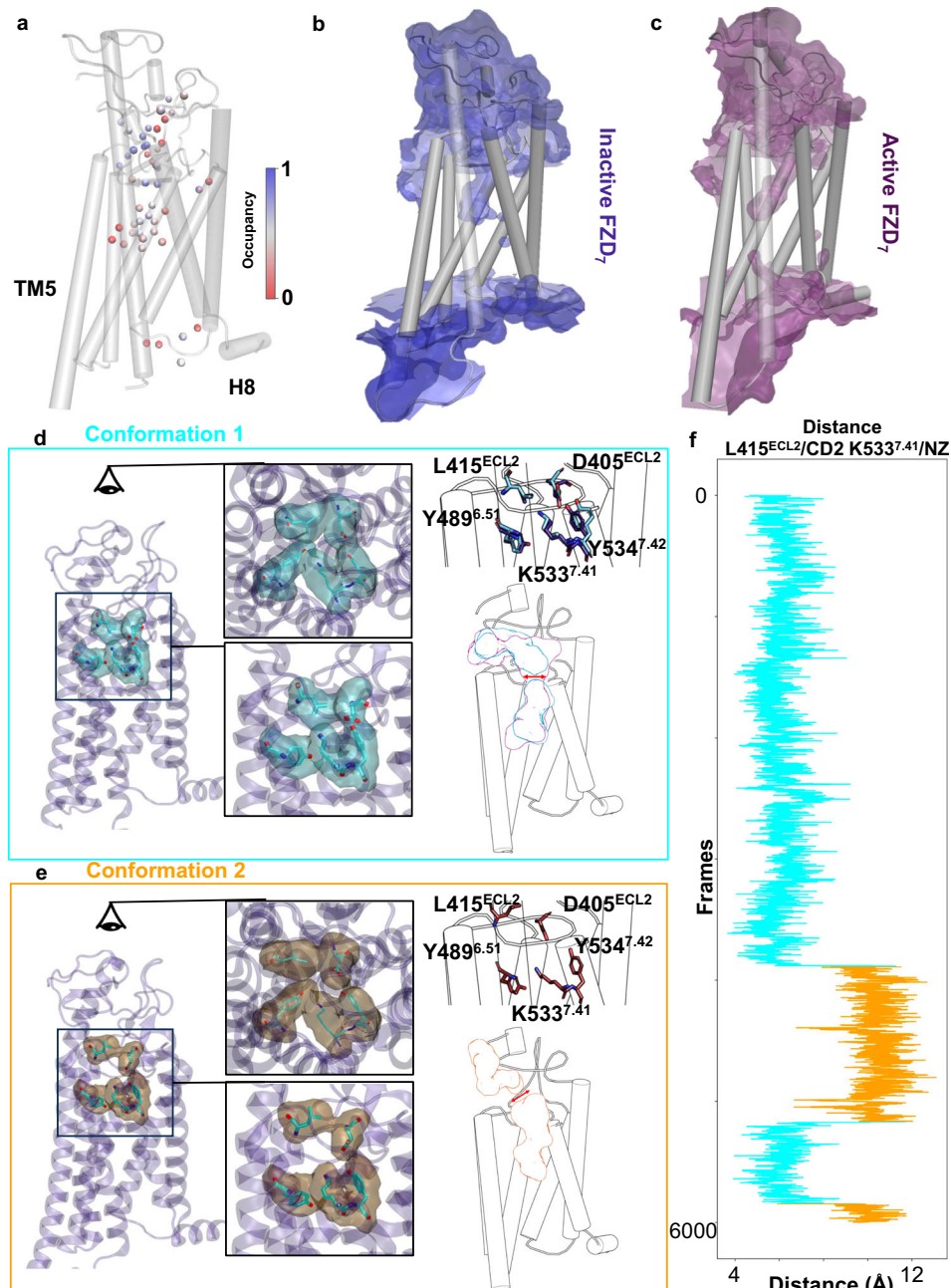

**Fig. 5 | Molecular dynamics of the water network in the inactive conformation of FZD_7. a** The occupancy and location of water molecules attributed to the internal water pocket with green depicting high occupancy and red depicting low occupancy during molecular dynamics simulations of the inactive FZD_7. Volumetric map for water molecules calculated for 20% occupancy with inactive **b** (blue density) and **c** active FZD_7 (purple density). **d** Key residues in the bottleneck region for the first conformation (transient bottleneck closing) with a volumetric map of the bottleneck residues computed with a threshold of 20% occupancy (blue density). **e** The second conformation represents a reorganization of ECL2 (disruption) with a volumetric map of the bottleneck residues that was computed with a threshold of 20% occupancy (brown density). **f** Distance calculations between L415^ECL2/CD2 and K533^7.41/NZ depict the two transient conformations shown in (**d** turquoise) and (**e** orange) over the course of 6000 frames of the MD simulations.

## FZD_7-binding cholesterol is critical for transducer association and signaling

Therefore, we structurally identified three residues that hydrophobically interact with cholesterol: F345^3.34, H382^4.46, and W386^4.50 (Figs. 6a and 7a, b). To better understand the functional implications of these interactions, we generated mutants where two of these three interacting residues were mutated to alanine (for a disruption of cholesterol-π (CH·π) contacts) (Figs. 6c and 7a, b). A NanoBiT-based approach was developed based on a C-terminally SmBiT-tagged

receptor (FZD_7-SmBiT or β_2-SmBiT) and a membrane-tethered LgBiT construct (FLAG-LgBiT-CAAX), which allows to selectively detect the fraction of receptors located at the plasma membrane without assay interference originating from intracellular, potentially immature receptors (Supplementary Fig. 9b, c). All generated mutants were validated for proper cell surface expression prior to probing function (Supplementary Fig. 9d–i).

The double mutants, F345^3.34A-H382^4.46A and F345^3.34A-W386^4.50A (Fig. 7a, b) were expressed at the cell surface level and adapted to our

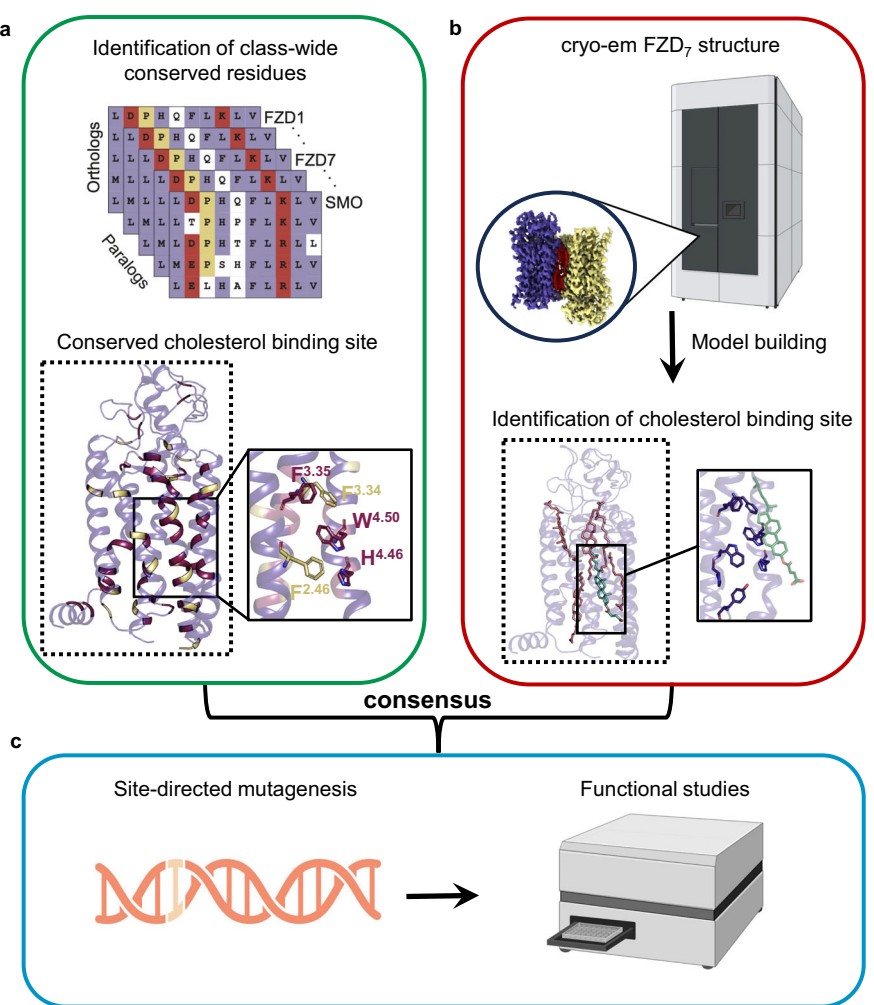

**Fig. 6 | Workflow for the investigation of a conserved cholesterol-binding site across class F GPCRs. a** A phylogenetic analysis was performed across class F GPCRs where highly conserved residues involved in cholesterol binding were identified. Fully conserved residues are colored in dark red and residues with similar conservation are colored in yellow. **b** Overview of the cryo-EM pipeline from data collection to model building and structural characterization of the cholesterol-binding site. **c** Based on the structural analysis, site-directed mutagenesis of cholesterol-interacting residues was performed and then used for analysis of functional downstream readouts of transducer coupling and signaling. Figure 6 was created with BioRender.com released under a Creative Commons Attribution-NonCommercial-NoDerivs 4.0 International license.

assay sensitivity (Supplementary Figs. 9c–i). We examined hetero-trimeric G protein activation by employing the previously published $G\alpha_s$ translocation assay (Supplementary Fig. 9a)[8]. Using pcDNA as control, we quantified the BRET between membrane-tethered rGFP-CAAX and $G\alpha_s$-67-Rluc in $\Delta FZD_{1-10}$ cells expressing $FZD_7$. Interestingly, the $FZD_7$ double mutants showed similar levels of constitutive activity towards $G_s$ compared to wild type $FZD_7$ highlighting the double mutant's full capability to functionally activate $G_s$ in a ligand-independent manner (Fig. 7c).

$FZD_7$ drives epithelial renewal by mediating WNT/β-catenin signaling, which is intrinsically dependent on the scaffold protein DVL. DVL interacts with FZDs through their DEP domain and serves as a hub for signalosome formation and signal initiation[13,41].

To further explore the mode of action of cholesterol on $FZD_7$, we tested the functionality of the mutants in a direct BRET assay monitoring Venus-DEP recruitment to $FZD_7$. Here, Venus served as the BRET acceptor and the wild type or mutant FZDs were C terminally tagged with SmBiT and served as the BRET donors (Supplementary Fig. 9b). The double mutants completely abrogated $FZD_7$-DEP recruitment, suggesting that cholesterol plays a key role in constitutive DVL recruitment by FZDs (Fig. 7d).

Furthermore, previous studies link WNT/β-catenin signaling to cholesterol[42], and suggest a direct link between dysregulation of cholesterol in the plasma membrane and aberrant WNT signaling[28]. Based on these findings, we ran a set of luminescence-based assays probing β-catenin-mediated regulation of T-cell factor/lymphoid enhancer factor (TCF/LEF) gene transcription (TOPFlash) to validate the impact of the mutations in the cholesterol-binding site of $FZD_7$ on WNT/β-catenin signaling. The $F345^{3.34}A$-$H382^{4.46}A$ and $F345^{3.34}A$-$W386^{4.50}A$ double mutations in $FZD_7$ reduced the WNT-3A-induced TOPFlash signal compared to wild type $FZD_7$, which is in line with their dramatically reduced ability to recruit DEP (Fig. 7e). These findings reveal a distinct mode of interaction between G protein and DVL with FZD, where cholesterol association to FZD is crucial for FZD-DVL interaction, while it is not required for constitutive G protein coupling.

## Discussion

Our results present high-resolution, structural insight into $FZD_7$ and the combination of cryo-EM structure, MD simulations, and phylogenetic analysis allows to draw FZD family-wide conclusions on structural aspects and mechanisms of FZD activation. These data are

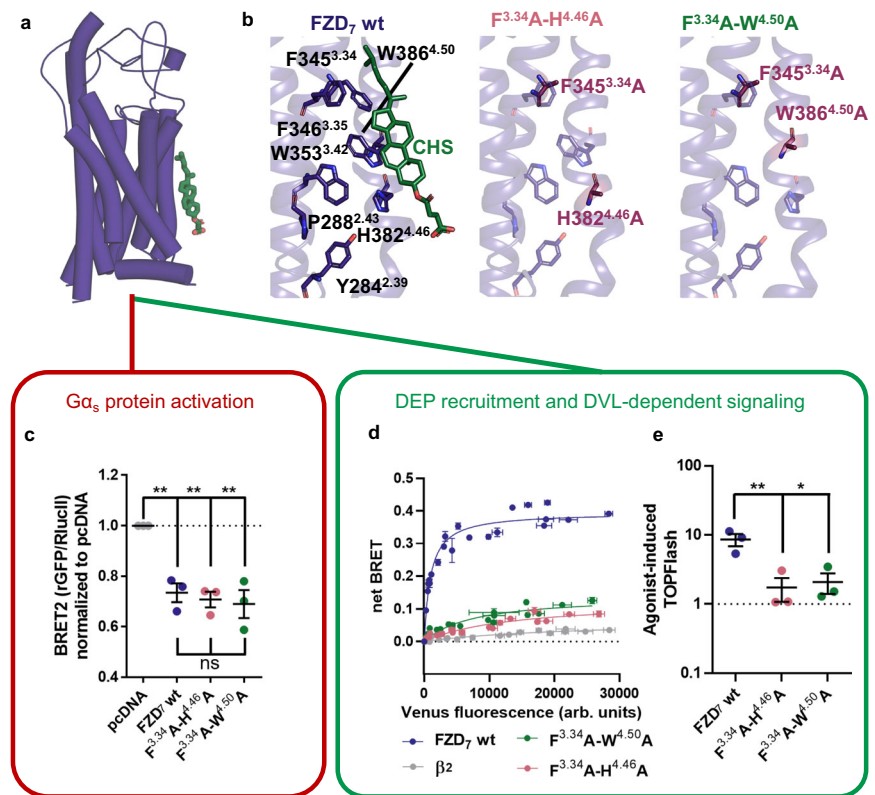

**Fig. 7 | Role of cholesterol in transducer coupling and signaling. a** Organization of the inactive FZD7 monomer with the cholesterol (CHS; green) binding site. **b** Analysis of the cholesterol-binding site reveals various interacting residues of which three were mutated for functional studies; F345$^{3.34}$, H382$^{4.46}$, and W386$^{4.50}$. **c** Gα$_s$ translocation assay showing BRET2 ratios of ΔFZD$_{1-10}$ HEK293T cells transiently transfected with rGFP-CAAX and Gα$_S$−67-RlucII, with either negative control (pcDNA), wild type FZD7, or the indicated FZD7 mutants normalized to conditions with pcDNA. Data are presented as means ± SEM of normalized BRET2 ratios from three independent experiments. (**$p < 0.001$, one-way ANOVA followed by Tukey's multiple comparison). **d** A NanoBiT-based DEP recruitment BRET assay was performed with FZD7-SmBiT, its respective cholesterol-binding site mutants, and β$_2$AR-SmBiT as a control in HEK293A cells. For functional reconstitution of Nluc,

membrane-anchored FLAG-LgBiT-CAAX was cotransfected. Data are presented in triplicates of three independent experiments with baseline-corrected BRET values based on conditions with 0% DEP-Venus. Data represent superimposed datapoints from three (FZD7wt and β$_2$) to four (FZD7 mutants) independent experiments, each performed in triplicate. Error bars in both x and y directions represent SD **e** ΔFZD$_{1-10}$ HEK293T cells were transfected with Renilla (Rluc), the Super 8x TOPFlash reporter, and wild type FZD7, or the indicated mutants. Cells were stimulated with 300 ng/mL of recombinant WNT-3A overnight. Data are presented as Fluc/Rluc ratios, normalized to the respective vehicle control, from three independent experiments performed in triplicates analyzed using one-way ANOVA (Dunnett post hoc test) (**$p$ 0.01, *$p < 0.05$). Data are presented as means ± SEM (error bars) from three independent experiments performed in triplicate.

complemented with experiments employing genetically encoded biosensors to functionally validate our structural findings. This integrative approach provides us with information to understand FZD7 function in particular and FZDs in general. The direct comparison between G protein-coupled and unbound FZD7 highlights that the conserved molecular switch (R/K$^{6.32}$-W$^{7.55}$) reported to be important for class F GPCR activation[7,23,32] is rearranged upon G protein association acting as hinge limiter, likely governing class F signaling. P$^{6.43}$ allows the TM6 to kink similarly to what is observed in class A GPCRs where the helix bend is accomplished by P$^{4.50}$ kink mediating G protein association[23,32]. In that regard FZDs differ from SMO, which lacks the proline and accommodates G protein association through parallel outward shift of TM6 instead of a kink. While it is now well established that FZDs are binding and activating heterotrimeric G proteins[7,8], systematic analysis of FZD mutants have unraveled that they tend to prefer DVL over G protein coupling suggesting that distinct conformational substates define transducer selectivity[5]. Interestingly, mutations affecting the molecular switch of FZD6, specifically R$^{6.32}$A and R$^{6.32}$Q/L, with R$^{6.32}$Q/L being naturally occurring cancer mutants of FZD6, show impaired recruitment of DVL. However, they enhance WNT-induced recruitment of miniG proteins to FZDs. This suggests that the positioning and dynamics of TM6-7 tightly regulate selectivity in coupling to DVL versus G protein[7].

Beyond validation of conformational changes upon transducer coupling, we shed light on two understudied receptor features, receptor-associated water molecules and cholesterol. The internal water pocket of the receptor and its plasticity highlight its potential for receptor dynamics on the extracellular side that could play a central role in agonist-induced allosteric conformational changes. This aspect is also mentioned in the reported structure of FZD4[22], but there are only a few structural waters identified in the electron density map, and the bottom of the pocket featured four non-attributed ions that are not present in the FZD7 structure. Additionally, the identification of a conserved, tight layer of residues forming the base of the internal pocket that maintains the same organization upon G protein association suggests that constitutive G protein coupling is not sufficient to propagate bidirectional allostery from the bottom to the top of the receptor in the absence of an agonist. Nonetheless, agonist stimulation of FZDs triggers a conformational change on the intracellular, transducer binding site of the receptor as shown both for G proteins as well as DVL[43] indicating that WNT-CRD interaction elicits conformational dynamics that are transferred to the transducer interaction site in an allosteric manner. This concept is further supported by diverse assessments of FZD dynamics such as the WNT-induced FZD-CRD dynamics[44] as well as agonist-induced FZD conformational changes[3,45]. On a structural level, however, it remains to be

elucidated how WNT stimulation of FZDs elicits full receptor activation and what conformational rearrangements are induced and required to promote DVL over G protein signaling.

We also identified a conserved cholesterol-binding site across class F GPCRs. In class A GPCRs cholesterol is mostly associated with structural stabilization[46] but can also be involved in signaling, as for example in the case of GPR161, where mutations that prevent cholesterol binding suppress $G_s$-mediated signaling, whereas other pathways remain unaffected[47].

In the case of SMO, cholesterol displays an agonist effect to mediate Hedgehog signaling. In this context, cholesterol targets internal binding sites and has been proposed to navigate in a narrow channel stretching from the bottom of TM5,6 to the receptor core/CRD to initiate receptor activation[48,49].

Previously, it was shown by a cholesterol-bead pulldown assay that cholesterol interacts with $FZD_5$ but not with other FZDs. Based on $FZD_5$ deletion mutants, the linker domain between the CRD and the FZD core was suggested to be involved in cholesterol binding and to affect $FZD_5$ function allosterically, for example in the context of pancreatic cancer[29]. These data stand in contrast to our findings regarding both the FZD paralog selectivity and the location of the cholesterol-binding site. However, the overall functional consequences of cholesterol on FZD maturation and signaling are similar since the cholesterol-site mutants of $FZD_7$ present with reduced DEP recruitment, reduced TOPFlash signal and reduced surface expression, translating to reduced ability to signal along the WNT/β-catenin pathway and impaired receptor maturation.

In conclusion, we provide a high-resolution structure of $FZD_7$ that allows family-wide conclusions on a FZD-specific receptor activation mechanism. Receptor activation comes along with a TM6 swing out at the same time as the molecular switch mechanism involving $R/K^{6.32}$ and $W^{7.55}$ acts as a hinge limiter dependent on the rotamer flip of $W^{7.55}$. This discovery defines a common mechanism of FZD activation providing a molecular explanation for the transducer selectivity of FZDs preferring DVL over G protein coupling. Furthermore, the high quality of our cryo-EM structure provided deeper insight into the dynamics of receptor-associated water molecules and a conserved, allosteric cholesterol-binding site that regulates DVL over G protein coupling. Both receptor-associated waters and the allosteric cholesterol site might open avenues for targeting FZDs pharmacologically by allosteric, small molecule compounds.

## Methods

### Figure preparation
Figures were designed using Pymol Molecular Graphics System (Schrödinger LLC) (v. 2.5) including the plugin CavitOmiX (v. 1.1.beta, 2024, Innophore GmbH); matplotlib (3.8.3), GraphPad Prism 10 (GraphPad Prism Software Inc.); ChimeraX (v. 1.5); VMD (v1.9.4.a55); biorender.com.

### Cells lines
HEK293A, (Thermofisher scientific)/R70507. ΔFZD_{1-10} HEK293T cells were kindly provided by Benoit Vanhollebeke. *Spodoptera frugiperda* (Sf9) insect cells (#11496015) are from Thermo Scientific.

### $FZD_7$ expression & purification
The full-length sequence of human $FZD_7$ was prepared from a HiBiT-$FZD_7$ construct (Addgene #195845) and cloned into a pFastBac1 vector containing a FLAG tag at the N-terminal, a 3 C cleavage site, and a Twin-Strep-tag® inserted on the C-terminal via Gibson cloning[50]. $FZD_7$ was expressed in *Spodoptera frugiperda* (Sf9) insect cells using the Bac-to-Bac baculovirus expression system (Thermo Fisher Scientific). Insect cells were grown in suspension in EX-CELL 420 Serum-free medium to a density of $2 \times 10^6$ cells/ml and infected with recombinant baculovirus

at a 1:50 v/v ratio. After culturing for approximately 48–54 h at 28 °C, cells were harvested by centrifugation and pellets were stored at −80 °C until use.

For the purification of $FZD_7$, cell pellets were thawed and lysed in buffer containing 10 mM TRIS-HCl pH 7.5, 100 mM NaCl, 1 mM EDTA, and protease inhibitors [leupeptin (5 μg/mL) (Sigma Aldrich), benzamidine (10 μg/mL) (Sigma) and phenylmethylsulfonyl (PMSF) (10 μg/mL) (Sigma Aldrich)]. After centrifugation (15 min at 3000 g), the pellet containing crude membranes was homogenized using a glass Dounce tissue grinder (10 strokes using A pestle then 20 strokes using B pestle) in a solubilization buffer containing 50 mM TRIS-HCl pH 8, 200 mM NaCl, 1% LMNG (Anatrace), 0.1% CHS (Sigma Aldrich), 0.1% GDN (Anatrace), iodacetamide (2 mg/mL) (Anatrace), and protease inhibitors. The mixture was stirred for 1.25 h at 4 °C and centrifuged (20 min at 38,400 g). The cleared supernatant was incubated with Strep-Tactin resin (IBA) for 2 h at 4 °C. The resin was washed with 10 column volumes (CV) of a buffer containing 50 mM TRIS-HCl pH 7.5, 500 mM NaCl, 0.02% lauryl maltose neopentyl glycol (LMNG), 0.002% cholesterol hemisuccinate tris salt (CHS), and 0.002% glycol-diosgenin (GDN). The resin was then washed with 15 column volumes (CV) of 50 mM TRIS-HCl pH 7.5, 100 mM NaCl, 0.02% LMNG, 0.002% CHS, and 0.002% GDN. $FZD_7$ was eluted in the same buffer supplemented with 2.5 mM desthiobiotin (IBA) and samples corresponding to the dimeric protein peaks ( ~ 8 mL) were concentrated in a 50-kDa molecular weight cutoff (MWCO) concentrator (Millipore) to 3.68 mg/mL.

After concentration, $FZD_7$ was loaded onto a size-exclusion chromatography column (SEC) (Superdex 200 Increase 10/300 GL, GE Healthcare) equilibrated with a buffer containing 10 mM TRIS-HCl pH 7.5, 100 mM NaCl, 0.002% LMNG, 0.0002% CHS, and 0.0002% GDN. Peak fractions corresponding to the dimeric receptor were flash-frozen and stored at −80 °C until further use.

### Cryo-EM sample preparation and image acquisition
The purified $FZD_7$-dimer fractions from the SEC were pooled and concentrated to 3.08 mg/mL (45 μM) in a 100-kDa MWCO concentrator. Montelukast was added at a molar ratio of 1:2 prior to grid freezing, however no small molecule was observed in the dataset. Next, a 3 μL sample was applied on glow-discharged (20 mA, 40 s.) UltrAuFoil R 1.2/1.3 300-mesh copper holey carbon grids (QuantiFoil, Micro Tools GmbH, Germany), blotted for 3.5 s, then flash-frozen in liquid ethane using Vitrobot Mark IV (Thermo Fisher Scientific). Images were collected on a Titan Krios G3i operating at 300 kV at the 3D-EM facility at Karolinska Institutet. Micrographs were recorded using a Gatan K3 detector in super-resolution mode using EPU software (v. 2.14.0). A total of 21,081 movies were obtained at a magnification of 165,000 corresponding to a 0.5076 Å calibrated pixel size and exposure dose of 80 e/Å$^2$ with defocus ranging from −0.6 μm to 2.0 μm (Supplementary Fig. 1).

### Cryo-EM data processing
Data processing for $FZD_7$-dimer was processed using cryoSPARC (v4.2-v4.4)[51,52]. Movie frames were aligned using Patch Motion Correction and Contrast Transfer Function (CTF) parameters were estimated by Patch CTF correction. Particle picking was performed by automatic Gaussian blob detection (mask diameter = 160 with elliptical and circular blob) yielding particles that were then subjected to reference-free 2D classifications (classes = 100, mask diameter = 160). Particles were extracted in a box size of 162 Å and downscaled to 2 Å/pixel. Iterative 2D classifications allowed the selection of 1,555,782 particles that were used as references to train a model with Topaz, a positive-unlabeled convolutional neural network for particle picking[53]. Topaz picked 7,332,734 particles that were used for further 2D classification. Particles were selected from the best 2D class averages (3,268,658

particles) and launched into 2 rounds of ab-initio model reconstructions ($C_1$) with 6 classes. Particles from the best class were reextracted with a pixel size of 0.6768 Å for further refinement. After NU-refinement with $C_2$ symmetry, the subset was subjected to global CTF refinement corrections for high order aberrations iterative refinement followed by two rounds of ab-initio model reconstructions ($C_1$) with two classes, ending up with a curated set of 105,380 particles. After NU-refinement, the final set was subjected to global CTF refinement corrections, reference-based motion correction, and then to a final NU-Refinement ($C_2$) yielding a map with 1.94 Å overall resolution (FSC 0.143) (Supplementary Fig. 1).

## Model building

Initially, the $FZD_7$–$mG_s$ (PDB: 7EVW) complex was used as a starting point for modeling. We first corrected the initial deposited model. To correct the model, we used the following tools including manual inspection and adjustments in Coot (0.9); global model refinement and relaxation in Rosetta (2022.45+release.20a5bfe), and global minimization with Phenix (1.20.1-4487) real-space refinement.

To build the inactive $FZD_7$ model, we started from the model of the $FZD_7$–$mG_s$ complex, extracted the receptor chain and applied the same strategy combining manual inspection and adjustments in Coot (0.9); global model refinement and relaxation in Rosetta, and global minimization with Phenix real space refinement. Overall statistics of the two final models are summarized (Supplementary Table 1).

## Plasmids and molecular cloning

The plasmid encoding DEP-Venus has been described previously[43]. Plasmids encoding $G\alpha_s$–67-$R$lucII and rGFP-CAAX were kindly provided by Prof. Michel Bouvier (IRIC, Université de Montréal, Canada). The TOPFlash reporter gene plasmid was from Addgene (#12456) and the plasmid for the constitutively expressed Renilla luciferase (pRL-TK) was from Promega.

C-terminally SmBiT-tagged receptor constructs ($FZD_7$-SmBiT, $\beta_2$-SmBiT) were generated in multiple steps. In the first step, the C-terminal Nluc tag in HA-$FZD_5$-Nluc[5] was exchanged with a SmBiT tag (resulting in HA-$FZD_5$-SmBiT) using the Q5 Site-Directed Mutagenesis Kit (New England Biolabs) according to the manufacturer's instructions. In a second step, the sequence for $FZD_5$ in HA-$FZD_5$-SmBiT was replaced with the nucleotide sequence for $FZD_7$ (amplified from HiBiT-$FZD_7$[50] and the $\beta_2$-adrenergic receptor (amplified from FLAG-SNAP-$\beta_2$ (kind gift from Davide Calebiro, University of Birmingham) via Gibson Assembly. Receptor mutants for $FZD_7$ were generated in the $FZD_7$-SmBiT backbone using the GeneArt Site-Directed Mutagenesis Kit (ThermoFisher) according to the manufacturer's instructions.

The membrane-anchored LgBiT construct (FLAG-LgBiT-CAAX) was generated based on DEP-Venus-kRas[43] via Gibson Assembly. The N-terminal FLAG tag was attached during the PCR by being implemented in the oligo design.

Generated plasmids were verified by Sanger Sequencing (Eurofins Genomics). A list containing all primers used to generate plasmids can be found in (Supplementary Table 4).

## Cell culture and transfection

HEK293A (human embryonic kidney cells) and $\Delta FZD_{1-10}$ HEK293T cells were used to functionally assess $FZD_7$ and the respective mutants with a various panel of biosensors. Cells were grown in Dulbecco's Modified Eagle's Medium (DMEM) supplemented with 2 mM glutamine, 10% fetal calf serum, 0.1 mg/mL streptomycin, and 100 units/mL penicillin and stored at 37 °C with 5% $CO_2$. Whenever indicated, cells were transfected in suspension with 1 µg of total DNA per mL cell suspension using linear polyethyleneimine (PEI Max, Polysciences Inc., stock concentration: 1 mg/mL) at a PEI:DNA ratio of 3:1. Transfected plasmid amounts indicated below always refer to the amount of transfected DNA per mL cell suspension.

## Confocal microscopy

HEK293A cells were seeded on a four-chamber 35 mm dish (ibidi, #80416) precoated with poly-D-lysine (PDL, Sigma Aldrich, #A3890401) at a density of 35,000 cells per quarter. The following day, cells were transfected with either FLAG-LgBiT-CAAX or empty pcDNA3.1 (negative control). After 24 h of incubation, cells were washed with PBS ( + $MgCl_2$, $CaCl_2$, Gibco #14080048, from now on only "PBS"), fixed with 4% paraformaldehyde (in PBS) and permeabilized using 0.25% Triton X-100 (ThermoFisher, #T8787) in PBS. After three washing steps with PBS, samples were blocked for 2 h with PBTA (1% bovine serum albumin (BSA), 0.05% Triton X-100 and 0.02% $NaN_3$ in PBS) and incubated overnight with anti-FLAG M2 antibody (Sigma Aldrich, #F1804, 1:1000 in PBTA) at 4 °C. After four washing steps with PBS and another blocking step for 30 min using PBTA, samples were incubated with a polyclonal goat anti-mouse secondary antibody (Alexa Fluor 488-conjugated, Invitrogen, #A28175, 1:1000 in PBTA) for 2 h at room temperature. Cells were washed four times with PBS, nuclei were counterstained with Hoechst 33342 (1 µg/mL) and cells were imaged in 0.1% BSA/HBSS using a Zeiss LSM800 confocal microscope.

## DEP recruitment assay

For DEP recruitment assays, 20 ng of $FZD_7$-SmBiT or 40 ng of $FZD_7$-SmBiT (mutants) or 20 ng of $\beta_2$-SmBiT (negative control) were transfected together with 300 ng of FLAG-LgBiT-CAAX and varying amounts of DEP-Venus (between 0 ng and 400 ng) into HEK293A cells (300,000 cells/mL, ad 1 µg DNA per mL cell suspension with pcDNA3.1). To assess the robustness of the assay setup with respect to variation in receptor membrane expression, a different set of experiments was performed using $FZD_7$-SmBiT wt in a range between 1 ng and 20 ng (Supplementary Figs. 8e-i). 100 µL/well of the transfected cells were seeded into PDL-coated, white 96-well plates. Cells were incubated for 48 h at 37 °C (with 5% $CO_2$) in a humidified incubator. Experiments were carried out using a TECAN Spark microplate reader. Cells were washed once with HBSS and maintained in 90 µL of HBSS. First, Venus fluorescence was measured (excitation 485/20 nm; emission 535/25 nm) followed by the addition of 10 µl of coelenterazine h (Biosynth, C-7004, final concentration: 5 µM final). After five minutes of incubation, Nanoluc (Nluc) luminescence (filtered between 445 and 485 nm) and the sensitized Venus emission (filtered between 520 and 560 nm) were detected. At least three independent experiments were conducted, and all conditions were run in triplicates. For data analysis, raw BRET/fluorescence was corrected by subtraction of the average value of the corresponding titration with 0% DEP-Venus. Net BRET data from the DEP recruitment assay were fitted using a one-site-specific binding equation that yielded $BRET_{50}$ and $BRET_{max}$ values. The corresponding $BRET_{50}$ values were log normalized to indicate a normal distribution. The plot of $BRET_{max}$ values over luminescence was fit using linear regression. Titration experiments were plotted and analyzed using GraphPad Prism 10 (GraphPad Prism Software Inc.).

## $G\alpha_s$ translocation assay

For the $G\alpha_s$ translocation assay, cells were transfected with 500 ng of the control plasmid, pcDNA3.1 (pcDNA) or $FZD_7$-wt (SmBiT-$FZD_7$) and its respective mutants, 25 ng $G\alpha_s$–67-$R$lucII, 300 ng rGFP-CAAX, and supplemented with pcDNA to yield 1 ug total DNA/mL cell suspension. Cells were mixed with the transfection reagent and 100 µL were seeded onto PDL-coated 96-well plates. Transfected cells were grown for 48 h at 37 °C with 5% $CO_2$ and washed once with HBSS. The substrate, coelenterazine 400a, was prepared at a final concentration of 2.5 µM and added to each well. Following incubation for 5 min, constitutive receptor activity was measured in a Tecan spark plate reader. $R$lucII emission intensity was quantified using a 400/40 monochromator and rGFP emission using a 540/35 monochromator with an integration time of 50 ms in both channels. The BRET2 ratio was defined as acceptor emission/donor emission or rGFP/$R$lucII and fitted using a

simple linear regression in Prism 5.0 software (GraphPad, San Diego, CA, USA). The BRET2 ratio was normalized to pcDNA levels and analyzed with one-way ANOVA and Tukey's multiple comparison post hoc test. Data are represented as means ± SEM from three independent experiments performed in triplicates (**$p < 0.001$).

## TOPFlash luciferase assay

$\Delta FZD_{1-10}$ cells HEK293 (450,000 cells/mL) were transfected in suspension with 100 ng of Nluc-$FZD_7$, 250 ng of the M50 Super 8x TOP-Flash reporter (Addgene #12456) and 50 ng of Renilla luciferase control plasmid (pRL-TK, Promega) per mL cell suspension. The control plasmid, pcDNA, was used to adjust the total transfected DNA amount of 1 µg per mL cell suspension. Cells were seeded (40,000 cells/well) onto a PDL-precoated, white-wall, white-bottomed 96-well microtiter plate (Thermo Fisher Scientific). After 24 h, cells were stimulated with either 300 ng/mL WNT-3A or vehicle control prepared in serum-free DMEM containing 10 nM of the porcupine inhibitor C59 (2-[4-(2-Methylpyridin-4-yl)phenyl]-N-[4-pyridin-3-yl)phenyl]acetamide, Abcam) to block secretion of endogenous WNTs.

About 24 h after stimulation, the Dual Luciferase Assay Kit (Promega, #E1910) was used for the readout. Briefly, cells were lysed with 20 µL of 1× Passive Lysis Buffer for 20 min at room temperature under shaking. Then 20 µL of LARII reagent was added to each well and β-catenin-dependent Fluc bioluminescence was detected using a Spark multimode microplate reader (Tecan, 550–620 nm, integration time: 2 s). Next, 20 µL of 1× Stop-and-Glo reagent was added to each well and Rluc bioluminescence was recorded (445-530 nm, integration time: 2 s) to account for differences in transfection efficiency. TOPFlash ratios were calculated by dividing the Fluc emission (β-catenin-dependent transcriptional activity) by the Rluc emission (indicator for transfection efficiency). Data are presented as means ± SEM from three independent experiments performed in triplicate. The significance level between wt $FZD_7$ and mutant $FZD_7$ was assessed by a one-way ANOVA, followed up by Dunnett's post-hoc test comparing all means to wt $FZD_7$, as implemented in GraphPad Prism10.

## Molecular dynamic simulations

$FZD_7$ (G protein bound (PDB:7EVW) and inactive) structures were used as a starting point for initial model generation. For inactive $FZD_7$, the sequence between residues 206-563 was used, the ICL3 (452-463) was added and refined in Coot with geometry restraints and without map restraints. For the $FZD_7$-$mG_s$ model, the sequence between residues 206-563 was used, while the sequence between residues 509 and 525 was added on the base of $FZD_7$ apo structure and refined in the same manner. A CHS molecule found in the conserved cholesterol-binding site was maintained in the system for both active and inactive $FZD_7$ and was parametrized using the CHARMM General Force Field and the CHARMM-GUI ligand reader.

The simulation system was generated in CHARMM-GUI bilayer builder from the online server (CHARMM-GUI)[54-63]. CHARMM36 is a standard force field used in molecular dynamics (MD) simulations to model the behavior of biological macromolecules, such as proteins, and is particularly well-suited for simulating membrane protein systems. It has been optimized to work with the TIP3P water model, a simple and widely used model in MD simulations, which represents a water molecule with three interaction sites: one for the oxygen and one for each hydrogen atom. While the simplicity of TIP3P can limit the accuracy of hydrogen bonding representation, more sophisticated models like TIP4P and TIP5P, which offer better representations of water's dielectric properties and hydrogen-bonding networks, can negatively affect protein behavior by altering the helix/coil equilibrium. Therefore, to ensure optimal simulation of the protein and observe water occupancy in the internal cavity without speculating on potential water-mediated hydrogen bonds, we adopted a conservative approach using the CHARMM36 force field with the TIP3P water model.

The receptors were pre-aligned in pymol (v. 2.5) on 7EVW (reference from Orientations of Proteins in Membranes (OPM) database (OPM (umich.edu)) and histidine protonations assignation was done manually in UCSF Chimera (v.1.13). The disulfide bonds: C210-C230; C234-C315; C336-C411; C508-515 were defined in CHARMM-GUI bilayer builder[56,64]. The receptors were capped with N-terminal acetyl and C-terminal CT3. A lipid bilayer with 100% palmitoyl-oleoyl-phosphatidylcholine (POPC) was generated and the system was solvated with TIP3 model and completed with 0.15 M $Na^+$ and $Cl^-$. Minimization, equilibration and productions runs were performed with the CHARMM36 and CHARMM36m force field in GROMACS(2023-2)[65].

The systems were first subjected to 5000-step minimization with the steepest descent algorithm integrator. The systems were further equilibrated in six successive steps with an iteratively decreasing force constant for the positional restraints for a total equilibration time of 22 ns and starting from randomly assigned velocities. The first initial two equilibration steps were run as NVT ensemble, before switching to the NPT ensemble for the remainder of the equilibration as well as production runs. Temperature coupling at 310 K was achieved using the v-rescale thermostat, while pressure coupling was achieved using the c-rescale barostat.

For the production runs, without positional restraints, random velocities based on the Boltzmann distribution were assigned to each of the 3 replicates for both systems and unbiased simulations were run for 300 ns with a time step of 2 fs. Hydrogen bonds were constrained using Linear constraints solver (LINCS)[66] and long-range electrostatic interactions were computed using Particle-Mesh Ewald (PME)[67] with 1.2 nm cutoff.

The full production trajectories were analyzed with both AMBER tool CPPTRAJ (V6.4.4)[68] calculating RMSD, dihedral angles and distances, and VMD (v1.9.4.a55)[69] using Volmap tool to compute volumetric maps based on water or residues occupancy. Prior to analysis, the trajectories were centered and aligned on the TMs with the respective software.

All trajectories are deposited on GPCRmd[70]. This simulation investigates fast dynamic events that don't require enhanced sampling as supported in similar studies[24,71].

## Identification of orthologs of human class F GPCRs

To be able to identify orthologous protein sequences we employed a multi-step pipeline. We used BLAST[72] (BLAST+ version 2.9.0) using the query human sequence of class F GPCRs against the UniProt database (retrieved at 14.07.2021). We fetched all sequences up until the 3rd human sequence hit and aligned them by using MAFFT FFTNS algorithm[73]. We trimmed the alignment by using clipkit-m kpic-gappy option[74] and generated a quick first tree by using FastTree with default parameters[74]. We retrieved the clade of the query protein by excluding the clades containing the other two human sequences. We realigned the identified subset of sequences and built multiple sequence alignment by using mafft linsi algorithm with maxiterate 1000 option. For tree reconstruction, we performed model selection with IQ-Tree 2[75]. We built trees with RaxML-NG 1.0.3 with the selected model, having removed diverged paralogs by using a previous algorithm described by ref. 76. The same paralogs were also removed from the multiple sequence alignment to calculate the residue conservation.

## Calculation of the residue conservation within orthologs

The most frequently observed amino acid was identified in a certain position with the following considerations: (i) If there is a gap, the residue is labeled as 0 conservation (ii) if it is not a gap, we continue our calculations. We calculated the total count of the most frequent amino acids with the total number of similar amino acid groups defined as the ones having a BLOSUM 80 score of >1. We divided the total number of identical and similar matches by the total number of

non-gap positions to calculate the percentage. If the percentage is equal to or greater than 90%, we labeled that position conserved.

## Alignment of human class F GPCRs sequences

We clustered the set of unfiltered sequences obtained using 0.65 identity by using cd-hit[77]. For each human receptor, we retrieved 5 representative sequences representing the largest calculated clusters. We aligned human and representative sequences of class F receptors by using mafft linsi algorithm. We removed the representative sequences and all gapped positions.

## Calculation of class-wide conservation and sequence variation in class F GPCRs

For each aligned position from the multiple sequence alignment of human sequences, we investigated if the homologous positions were previously labeled as conserved. To calculate the conservation of the position across paralogs we used the percentage of receptors having a conserved amino acid as was previously described above. For example, if 8 out of 11 receptors are identified as conserved at a specific residue, we calculate the conservation as 8/11 or 72.7% Residue divergence is calculated by comparing the most frequently observed residues for each receptor to each other. We retrieved this information from all 11 class F GPCRs and used entropy as a measure of sequence divergence. For the conserved water network analysis, positions with 0 entropy were used (all identical residues in class F GPCRs) and 0.44 entropy (one of the receptors in the class is different). For the cholesterol analysis, positions with 0 entropy and positions that only showed strict similarity based on BLOSUM 80 score were used.

## Calculation of the residue-residue contact score change network

We applied the previously published residue-residue contact score (RRCS)[78] algorithm to inactive and active state $FZD_7$ structures using a custom Python script to subtract $RRCS_{inactive}$ from $RRCS_{active}$ to identify the changes observed during the activation of the receptor.

## Reporting summary

Further information on research design is available in the Nature Portfolio Reporting Summary linked to this article.

## Data availability

The cryo-EM density maps for $FZD_7$ dimer have been deposited in the Electron Microscopy Data Bank (EMDB) under accession codes EMD-19881. The coordinates for the models of the amended $FZD_7$-Gs and for $FZD_7$ dimer have been respectively deposited in the PDB under accession numbers 9EW2 and 9EPO. All molecular dynamic trajectories were deposited on GPCRmd (inactive $FZD_7$ ID: 2064 and active $FZD_7$ ID: 2065). Source data are provided with this paper.

## Code availability

Multiple sequence alignments of paralogs and orthologs used for evolutionary analysis, and code and input files used for residue-residue contact analysis are provided at https://github.com/CompGenomeLab/fzd7_evolution_and_structure. https://doi.org/10.5281/zenodo.13175921.

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

## Acknowledgements

We thank Pankonin Maik for his advice and discussions regarding the analysis of the molecular dynamics simulations. We thank Rémy Sounier for his contribution to figure design. Thanks to Prof Michel Bouvier for sharing relevant plasmids. EM data were collected at the Karolinska Institutet 3D-EM facility https://ki.se/cmb/3d-em. Samples were screened in SciLifeLab for access to the cryo-EM Swedish National Facility. The work was supported by grants from Karolinska Institutet, the partial funding to doctoral students at Karolinska Institutet (2021-00430), the Swedish Research Council (GS: 2019-01190), the Swedish Cancer Society (GS: 20 1102 PjF, 23 2825 Pj), the Novo Nordisk Foundation (GS: NNF22OC0078104), The German Research Foundation (DFG; LG: 504098926; JHV: 520506488), Svenska Sällskapet för Medicinsk Forskning, SSMF (MMS: PG-22-0379; JB: PG-23-0321) and the Wenner Gren Foundations (UPD2021-0029). The MD simulations were enabled by resources provided by the National Academic Infrastructure for Supercomputing in Sweden (NAISS; NAISS 2023/5-419), partially funded by the Swedish Research Council through grant agreement no. 2022-06725.

## Author contributions

G.S., J.B., and J.K. initiated and designed the project. J.B., and J.K. carried out the FZD₇ purification and cryo-EM acquisition and analysis. J.B., and J.K. built the model and interpreted it. J.B., and M.S., ran the molecular dynamic analysis and interpreted the results. B.S., and O.A. Designed a strategy for the phylogenetic analysis of class F GPCRs and ran the analysis. L.G., and J.B. designed a rational strategy for cholesterol-binding site mutations. L.G., J.K., JV., and J.B. performed the wet lab experiments. J.B., L.G., J.K., B.S., O.A., and G.S. designed and prepared the figures. J.B., J.K., and G.S. wrote the manuscript. B.S., and O.A. wrote the phylogenetic analysis sections L.G., J.V., B.S., and O.A. commented and contributed to the manuscript writing. G.S. supervised and coordinated the project.

## Funding

## Competing interests

The authors declare no competing interests.
