## [Peer Review File · Nature Communications]

Reviewers' Comments:

Reviewer #1:

Remarks to the Author:

Bous et al. from Gunnar Schulte's group reported the high-resolution apo structure of FZD7. The high resolution was obtained in a rather surprising way, as FZD7 forms an antiparallel dimer with C2 symmetry. Due to the high resolution, some waters, lipids, and cholesterol were observed. The authors claimed that the antiparallel dimer was 'artificial'. From this 'artificial' interface, functional FZD7 cholesterol binding sites were derived, and this may be the main point of novelty.

Major concerns:

The authors labelled the observed antiparallel dimer as 'artificial', although no deliberate steps were taken to artificially induce its formation. The protein extraction process followed standard procedures using normal detergents. One notable exception was the addition of the cytochrome P450 inhibitor Montelukast. However, the rationale behind this addition was not explained, and there was no discussion on the potential contribution of Montelukast to this dimerization. Furthermore, no control dataset without Montelukast was provided. This observed antiparallel dimer contradicts their previous publication on FZD6, where a parallel dimerizing interface involving TM4/5 was well characterized. Understanding the dimerization status of FZD receptors is crucial for Wnt signalling. Given that the authors have studied both FZD7 and FZD6, it raises the question of whether there are differences in dimerization orientation between the two receptors.

For high-resolution structures, it is not surprising that water densities can be observed. However, over-interpretation of these waters may not be informative. The authors analysed both active and inactive FZD7 waters. However, is the previously published resolution of active FZD7 (3.2Å) sufficient for water analysis?

Cholesterol binds to F-class receptors, as exemplified by SMO, where the binding sites within the core of the protein are very important for receptor function. The possibility of FZDs binding with cholesterol has been thoroughly investigated using a well-defined chemical approach (as referenced in #24), concluding that FZD7 does not show any signs of cholesterol binding, whereas FZD5 (and possibly FZD8) could bind to cholesterol. The proposed cholesterol binding sites (identified by mutational analysis) for FZD5 are within the core of the receptor. In this study, the authors characterized cholesterol binding sites outside of the FZD7 core protein, specifically from the 'artificial' interface. It is crucial to have experimental evidence to show that FZD7 actually binds to cholesterol. In high-resolution membrane structures, cholesterol

hemisuccinate/lipids are frequently observed outside of core proteins, but these sites tend not to be functional. For example, WLS-Wnt3 at a resolution of 2.2Å (PDB 7DRT) or those seen in the dimer interface (PDB 4OR2)

Reviewer #2:

Remarks to the Author:

In their manuscript, "Structural Basis of Frizzled 7 Activation and Allosteric Regulation," Bous et al. present the first inactive-state structure of the FZD7 receptor. This extremely high-resolution structure (1.9Å) for the first time allowed investigations into cholesterol binding at FZD receptors, identified the water network facilitating receptor activation, and the mechanisms underlying this process. This work offers a direct comparative analysis with the previously determined (and now re-refined) active-state model of the FZD7-mGs complex, revealing substantial conformational shifts at the transmembrane level and pinpointing the microswitches directly responsible for receptor activation. This is the first study to directly show the mechanisms responsible for FZD receptor activation and G protein coupling.

A notable innovation in this study is the absence of a stabilising agent, commonly used in GPCR structural biology, to "bulk up" receptors for inactive state investigation by cryo-EM. Instead, the researchers cleverly utilised the antiparallel dimers observed during purification. Although the dimers are non-physiological, they provide a novel approach for stabilising receptors in their inactive state, enabling the acquisition of very high-resolution structures of inactive GPCRs. I congratulate the authors for this remarkable achievement! While the potential applicability of this method for structure determination of other GPCRs in inactive states remains uncertain and warrants further investigation, the possibility is very intriguing.

The authors also explored the implications of cholesterol binding on FZD7, and other FZD receptors. The cholesterol binding is well-known in more common Class A and B GPCRs where cholesterol molecules are often observed in high-resolution structures. The authors identified a conserved cholesterol-binding site across FZD receptors, which may play a role in signalling. Intriguingly, cholesterol site mutants exhibit distinct impacts on G protein- and Dvl-mediated signalling. Further investigation into this phenomenon could provide valuable insights. Specifically, it would be interesting to explore whether these differences stem from ligand binding effects (Dvl coupling) versus constitutive activity (G protein coupling).

Another potential explanation for the observed reduction in TOPFlash signalling could be the interference of mutants with the co-receptor (LRP5/6) engagement. While the diminished DEP binding likely eliminates the chance that the mutant effect is mediated via LRP5/6, further delineating cholesterol effects vs. co-receptor interactions would benefit this study. Additionally, experiments investigating the impact of cholesterol

depletion, such as using methyl- β -cyclodextrin, could provide even deeper insights. Overall, this is an important and impactful study that sheds light on the mechanism of receptor activation for understudied and clinically relevant FZD GPCRs.

Minor comments/questions:

1. Figure 4A shows that RMSDs for 3 independent simulations are diverging; where does this come from? I.e. what is the most flexible (divergent) areas of FZD7 receptor in the simulation?
2. Line 188. “the difference in TM6 dynamics” – it appears you are talking about the extent of the bend evident from the structures, but rather not dynamics per se
3. Figure 1C - what is the reason for surface representations of the ECL loops and the hinge.
4. Figure 1E - the colours are really hard to see
5. Figure 2H and I – is this the angle distribution across multiple time points in the simulation? please provide more information in the legend
6. Figure 5A-C colours are really hard to see

Reviewer #3:

Remarks to the Author:

This paper provides insight on the FZD7 receptor and allosteric regulation. As I am a computational biophysicist, I will comment mostly on the molecular dynamics (MD) simulations.

- The MD simulations were carried out in the NVT ensemble, but we are not told what thermostat has been used. Why not running NPT simulations?

- The MD was used to investigate the distribution of water molecules around the protein. How much this distribution depends on the force field used for the water model?

-More references about F GPCRS besides ref. 1 (which is from some of the authors) could be added.

-There are repetitions in the paper ('The most extensively studied GPCR classes, A, B, and C, contain conserved motifs, allosteric sodium binding sites, and cholesterol binding sites as well as an intricate network of waters that define the dynamic conformational landscape of the receptor. In the case for FZDs, these features remain to be elucidated.' and 'Frizzled ...are class F GPCRs that remain understudied')

-

Reviewer comments – rebuttal letter

We are thankful for the overall positive reception and appreciation of our work. The reviewers provided constructive criticism, which helped us to revise and improve our manuscript.

(reviewer comments in GREEN; author responses in **black**)

REVIEWER COMMENTS

Reviewer #1 (Remarks to the Author):

Bous et al. from Gunnar Schulte's group reported the high-resolution apo structure of FZD7. The high resolution was obtained in a rather surprising way, as FZD7 forms an antiparallel dimer with C2 symmetry. Due to the high resolution, some waters, lipids, and cholesterols were observed. The authors claimed that the antiparallel dimer was 'artificial'. From this 'artificial' interface, functional FZD7 cholesterol binding sites were derived, and this may be the main point of novelty.

Response: We appreciate the constructive comments of reviewer #1 and provide clarification on the dimerization of FZD₇ in this context and the reasons why we assume that it is an artificial dimer that will not occur in physiologically relevant systems as well as clarification on a few other key points.

Major concerns:

The authors labelled the observed antiparallel dimer as 'artificial', although no deliberate steps were taken to artificially induce its formation. The protein extraction process followed standard procedures using normal detergents.

Response: While it is noteworthy that specific seven transmembrane-spanning proteins (i.e.: ceramidase) feature an inverted topology respectively to GPCRs, it is important to highlight that the orientation of GPCRs is tightly regulated with the N-terminal domain oriented towards the extracellular surface and the C-terminal domain facing the intracellular side. This is determined by the signal N-terminal signal peptide and this orientation is essential for ligand binding and receptor signaling. Thus, antiparallel GPCRs do not occur naturally in cells. To our knowledge, there is no experimental evidence pointing into this direction. Moreover, this aspect is not part of the scope of our work. Therefore, we hypothesize the antiparallel dimers are formed during the purification process and

more specifically during solubilization. Under these conditions, the antiparallel dimer likely represents a more thermodynamically stable entity compared to monomers or parallel dimers. Furthermore, we utilize protein overexpression with Sf9 cells, which is an ideal model system for expression of recombinant GPCRs and shown to have similar receptor properties in mammalian cells (1). This system allows for high levels of expression, which upon solubilization, could favor the formation of antiparallel dimers. Thus, these conditions cannot occur naturally in cells due to the strict “N-terminus out” orientation in the plasma membrane essential for ligand binding, subsequent signaling and transducer coupling. Other nonclassical Cryo-EM structures of antiparallel dimers were characterized before as shown in the two following examples (2, 3).

This observed antiparallel dimer contradicts their previous publication on FZD6, where a parallel dimerizing interface involving TM4/5 was well characterized.

Understanding the dimerization status of FZD receptors is crucial for Wnt signalling. Given that the authors have studied both FZD7 and FZD6, it raises the question of whether there are differences in dimerization orientation between the two receptors.

Response: First of all, we would again like to underline that the observed, non-physiological antiparallel dimer has nothing to do with the role of FZD dimerization for cellular signaling. In fact, it would be very misleading to draw any conclusions from one scenario to the other.

Nevertheless, we agree with the reviewer that FZD dimerization is physiologically important. We and others have studied the role and dynamics of FZD dimerization for example in the case of FZD_{4, 5, 6, 7} (4–7), the need to understand the dimerization status in regard to its potential impact on signaling, for example in the case of FZD₆ (4). Again, we would like to highlight that these studies of dimerization in living cells cannot be compared to the scenario in the detergent-solubilized purified receptor.

One notable exception was the addition of the cytochrome P450 inhibitor Montelukast. However, the rationale behind this addition was not explained, and there was no discussion on the potential contribution of Montelukast to this dimerization. Furthermore, no control dataset without Montelukast was provided.

Response: The addition of montelukast has no effect on the formation of antiparallel dimers or the overall receptor structure. Montelukast was added after solubilization and the formation of dimers and their subsequent purification. The conditions that included montelukast represented an attempt to detect potential interaction between montelukast and FZD₇. However, we were not able to identify a density for montelukast in the ensemble of structures presented here. Importantly, we solved two CryoEM structures of the antiparallel FZD₇ dimer, one in the

presence and one in the absence of montelukast. Also, the two datasets were acquired on two different microscopes with different setups and conditions. The different data sets led to basically identical densities but different resolutions (1.9 vs 2.4 Å). Therefore, we continued with the dataset that reached a higher resolution, which was the FZD₇ structure with the addition of montelukast. The high-resolution obtained provided us the opportunity to deduce more information including the water network analysis and the identification of receptor-associated lipids.

For high-resolution structures, it is not surprising that water densities can be observed. However, over-interpretation of these waters may not be informative. The authors analysed both active and inactive FZD₇ waters. However, is the previously published resolution of active FZD₇ (3.2Å) sufficient for water analysis?

Response: We strongly agree with reviewer1 and therefore we did not build any water molecules in the re-refined FZD₇-mGs based on the Cryo-EM density. We

investigated the distribution of water molecules in a Molecular Dynamic setup that is not derived from any interpretation of water location from the FZD₇-mGs Cryo-EM map and observed the presence of waters and their dynamics in the internal cavity during the simulation.

Cholesterol binds to F-class receptors, as exemplified by SMO, where the binding sites within the core of the protein are very important for receptor function. The possibility of FZDs binding with cholesterol has been thoroughly investigated using a well-defined chemical approach (as referenced in #24), concluding that FZD7 does not show any signs of cholesterol binding, whereas FZD5 (and possibly FZD8) could bind to cholesterol. The proposed cholesterol binding sites (identified by mutational analysis) for FZD5 are within the core of the receptor. In this study, the authors characterized cholesterol binding sites outside of the FZD7 core protein, specifically from the 'artificial' interface. It is crucial to have experimental evidence to show that FZD7 actually binds to cholesterol. In high-resolution membrane structures, cholesteryl hemisuccinate/lipids are frequently observed outside of core proteins, but these sites tend not to be functional. For example, WLS-Wnt3 at a resolution of 2.2Å (PDB 7DRT) or those seen in the dimer interface (PDB 4OR2)

Response: As stated by Reviewer #1 SMO is a striking example of Class F GPCRs binding to cholesterol with functional relevance. It is worth to note that in SMO, TM6 opening is not generated by a kink in the TM6 but rather by a parallel outward shift of TM6 resulting in a lateral opening of a hydrophobic tunnel where cholesterol was proposed to navigate (8). Compared to SMO, mechanisms of FZD activation are rather different with the opening of TM6 induced by a kink of P^{6.43} without the formation of a hydrophobic tunnel (8). To our knowledge, there is no functionally defined and validated evidence of cholesterol interaction with FZD in the 7TM core, neither within the helical barrel nor around.

Regarding the lack of evidence in the literature suggesting a role of cholesterol for FZD₇ signaling: While reference #24 provides compelling evidence for the presence of a cholesterol binding site located in the FZD₅ linker/ECL1 domain (not present in FZD₇), the applied methodology restricts firm conclusions on potential binding sites in the membrane section, because cell lysis in 1% NP40 and the use of NP-40 for successive washes could easily shed physiologically relevant lipids associated with the membrane section resulting in dissociation of FZD₇ from the beads.

Furthermore, the following publication (9) highlights the importance of cholesterol for β-catenin signaling in the context of colorectal cancers that involved FZD₇ and not FZD₅ and shows a reduced FZD₇-LRP and FZD₇-DEP FRET signal in membranes depleted of cholesterol among other findings (9). While these analyses lend our work substantial physiological relevance, we extend the molecular resolution of those concepts by pinpointing a conserved and functionally important cholesterol interaction site in the 7TM core of FZDs.

As stated by reviewer #1, we identified cholesterol binding sites outside of the FZD₇ core protein, in the 'artificial' interface, that required careful validation that is detailed in the manuscript. First, as stated in the manuscript, a parallel, independent, and unbiased phylogenetic analysis was carried out by Oğün Adebali and Berkay Selcuk, which led to the unambiguous identification of the very same conserved cholesterol binding site that was discovered in the CryoEM structure. To validate these data further, we used site-directed mutagenesis to alter the cholesterol binding site, and observed a functional effect emphasizing that this site is indeed important for downstream signaling unlike what is suggested by reviewer #1. While it is usually rather challenging to clearly demonstrate that site-directed mutagenesis can impact the cholesterol binding and not the protein folding and functionality, in our case the mutants conserve their ability to constitutively recruit heterotrimeric Gs proteins. Altogether, this demonstrates that the mutants are properly folded and addressed for G protein coupling, while they abrogate DEP recruitment and subsequently also WNT/ β -catenin signaling. Collectively, our results provide strong evidence for a functional and conserved cholesterol binding site in FZD₇.

Reviewer #2 (Remarks to the Author):

In their manuscript, "Structural Basis of Frizzled 7 Activation and Allosteric Regulation," Bous et al. present the first inactive-state structure of the FZD7 receptor. This extremely high-resolution structure (1.9Å) for the first time allowed investigations into cholesterol binding at FZD receptors, identified the water network facilitating receptor activation, and the mechanisms underlying this process. This work offers a direct comparative analysis with the previously determined (and now re-refined) active-state model of the FZD7-mGs complex, revealing substantial conformational shifts at the transmembrane level and pinpointing the microswitches directly responsible for receptor activation. This is the first study to directly show the mechanisms responsible for FZD receptor activation and G protein coupling.

A notable innovation in this study is the absence of a stabilising agent, commonly used in GPCR structural biology, to "bulk up" receptors for inactive state investigation by cryo-EM. Instead, the researchers cleverly utilised the antiparallel dimers observed during purification. Although the dimers are non-physiological, they provide a novel approach for stabilising receptors in their inactive state, enabling the acquisition of very high-resolution structures of inactive GPCRs. I congratulate the authors for this remarkable achievement! While the potential applicability of this method for structure determination of other GPCRs in inactive states remains uncertain and warrants further investigation, the possibility is very intriguing.

Response: Thank you for your thoughtful review. We highly appreciate your

positive feedback on our manuscript and that you highlight the interest of using the antiparallel dimers instead of conventional stabilization strategies to solve FZD₇ in an inactive conformation. Indeed, while these dimers are often disregarded, they provide a great opportunity for structural investigation. Like Reviewer #2 rightfully states, the applicability of this strategy to other GPCRs is limited by the capacity to form such stable dimers. Interestingly similar approaches were used in at least two similar cases (2, 3).

The authors also explored the implications of cholesterol binding on FZD₇, and other FZD receptors. The cholesterol binding is well-known in more common Class A and B GPCRs where cholesterol molecules are often observed in high-resolution structures. The authors identified a conserved cholesterol-binding site across FZD receptors, which may play a role in signalling. Intriguingly, cholesterol site mutants exhibit distinct impacts on G protein- and Dvl-mediated signalling. Further investigation into this phenomenon could provide valuable insights. Specifically, it would be interesting to explore whether these differences stem from ligand binding effects (Dvl coupling) versus constitutive activity (G protein coupling).

Response: Reviewer #2 makes an excellent point regarding the investigation of DVL coupling of the mutants under agonist stimulation. We acknowledge the interest in this area. However, it is important to note that the interaction between FZDs and DVL (or the isolated DEP domain) occurs even in the absence of agonists, at least under overexpression conditions. This interaction is somehow similar to constitutive coupling to heterotrimeric Gs. While agonist-induced rearrangements in the FZD-DVL interface are not fully understood (as discussed in our recent paper on this topic – (10)), it would indeed be valuable to study the role of FZD-cholesterol binding in FZD₇-DVL recruitment and rearrangement under stimulation.

To address this, we conducted a suitable experiment (see below). However, due to the limited sensitivity of the setup and the relatively low surface expression of the mutant receptor (Supp Fig 8E), our interpretation is limited. We cannot confidently claim that the absence of a change in BRET in response to WNTs is caused by the mutants interfering with the FZD-DEP interaction properties, rather than a limitation of the assay's sensitivity. It should be underlined that WNT-16B shows a distinct Δ BRET response in corresponding assays with wild type FZD₇. Consequently, we decided to not include it into the manuscript.

FZD₇-SmBiT+LgBiT-CAAX+DEP-Venus WNT stimulation

Another potential explanation for the observed reduction in TOPFlash signalling could be the interference of mutants with the co-receptor (LRP5/6) engagement. While the diminished DEP binding likely eliminates the chance that the mutant effect is mediated via LRP5/6, further delineating cholesterol effects vs. co-receptor interactions would benefit this study.

Response: First of all, we surely agree with the reviewer's statement that reduced DEP binding eliminates "the chance that the mutant effect is mediated via LRP5/6". To our knowledge, the main complexing force in the FZD-LRP5/6 interaction is the WNT-mediated bridging of the ligand binding domains rather than interaction of the transmembrane domains of these receptors. Nevertheless, the question is valid and indeed interesting to pursue. So far, our attempts to probe this impact through a direct FZD7-LRP5/6 BRET approach were unsuccessful (see below) because the low cell surface expression of the mutants (Supp Fig 8E) is not sufficient regarding the sensitivity of our FZD7-LRP5/6 assay preventing to draw clear conclusions on the mutants/LRP5/6 interaction.

There is nonetheless evidence in (9) that a role of cholesterol in the FZD-LRP interaction as assessed by FRET in the absence and presence of WNT/MβCD:

[Editorial note: this figure was redacted due to third-party rights. It can be found in [3], Figure 7c].

However, the experimental paradigm does not really allow firm conclusion on what is affected by MβCD treatment as several parameters could be possible (membrane fluidity, protein compartmentation, protein mobility, effects on WNTs and WNT-receptor interaction etc). It is difficult to extract that cholesterol binding to FZD is the only factor that explains the changes in the presence of MβCD.

Additionally, experiments investigating the impact of cholesterol depletion, such as using methyl-β-cyclodextrin, could provide even deeper insights.

Response: While it is noteworthy that MβCD has been previously used in colorectal cancer systems to study the effects of cholesterol depletion on β-catenin signaling, particularly focusing on the formation of the signalosome and the colocalization of signalosome proteins (FZD₇, DVL, LRP6), the results indicate that cholesterol

depletion does alter signalosome formation (9). However, M β CD significantly impacts overall cell viability, membrane fluidity, and protein mobility and stability.

In our experiments, using M β CD resulted in a dramatic decrease in cell viability. Given these broad and severe effects of cholesterol depletion, we are not confident that it directly impacts β -catenin signaling. Therefore, we believe that M β CD treatment does not provide clear evidence for FZD-cholesterol binding, nor does it help to demonstrate the functional and mechanistic relevance of the conserved cholesterol-binding site on FZDs.

Minor comments/questions:

1. Figure 4A shows that RMSDs for 3 independent simulations are diverging; where does this come from? I.e. what is the most flexible (divergent) areas of FZD7 receptor in the simulation?

Response: After a careful inspection, we discovered an error in the backbone RMSD plot for Production 1, which explains the observed divergence. We apologize for this mistake and have replotted the corresponding panel. To further illustrate how residues influence RMSD, we added a B-factor plot by residues for the different simulation replicates, as shown in Supplementary Figure 5. Interestingly, higher RMSD values in the inactive FZD₇ MD are caused by dynamics in ICL3, which partially fold as an α -helix in the active conformation, thereby restricting its flexibility.

2. Line 188. "the difference in TM6 dynamics" – it appears you are talking about the extent of the bend evident from the structures, but rather not dynamics per se

Response: In Line 188 "Interestingly, the difference in TM6 dynamics between class A (β_2 AR) (Fig. 3A) and class B1 GCGR (Fig. 3B) GPCRs is correlated to their respective efficacy to couple to and activate the heterotrimeric G protein with GCGR exhibiting a substantially lower guanine nucleotide exchange activity (11)." Actually, refers to the dynamics of TM6 in class A and B as described in the referred paper (11). This highlights the importance of TM6 in G protein binding behavior consistent with the idea that Class F TM6 limited opening is suboptimal for G protein binding. From ref (11):

[Editorial note: this figure was redacted due to third-party rights. It can be found in [3], Figure 8].

3. Figure 1C - what is the reason for surface representations of the ECL loops and the hinge.

Response: ECL loops representation as a surface give a better idea on the position of the peripheral lid and to which extent it obtrudes access to the receptor core.

4. Figure 1E - the colours are really hard to see

Response: The figure was redesigned to improve visualization with the receptor in grey and a slight transparency.

5. Figure 2H and I – is this the angle distribution across multiple time points in the simulation? please provide more information in the legend

Response: Yes, angle distribution across multiple time points (time step 50ps) were plotted for the three replicates using gnuplot with consistent results. The plots displayed in figure 2h and i represent the results for replicate 1 of the active and inactive states MD. (plots for the other replicas were included Supplementary Figure 5)

The figure legend was completed as followed "Scatterplot of occurring χ_1 χ_2 dihedral angles of residue (H) W^{3×43} and (I) W547^{7.55} calculated for each frame (50 ps time steps) over the trajectory from replica 1 of FZD₇ simulations of active and inactive state. "

6. Figure 5A-C colours are really hard to see

Response: The figure was redesigned to improve visualization.

Reviewer #3 (Remarks to the Author):

This paper provides insight on the FZD7 receptor and allosteric regulation. As I am a computational biophysics, I will comment mostly on the molecular dynamics (MD) simulations.

Response: We thank Reviewer #3 for the assessment of molecular dynamics simulations in our manuscript highlighting the need to develop further the MD method section as well as raising an interesting point about the impact of the force field on the water analysis.

- The MD simulations were carried out in the NVT ensemble, but we are not told what thermostat has been used. Why not running NPT simulations?

Response: We apologize for the confusion in the method section. Only the first two steps of the equilibration were run in an NVT ensemble, while the remaining equilibration as well as production runs were run as NPT ensemble using v-rescale thermostat and c-rescale barostat. The method section has been modified as follow to improve clarity: "The systems were first subjected to 5000 steps minimization with the steepest descent algorithm integrator. The systems were further equilibrated in six successive steps with an iteratively decreasing force constant for the positional restraints for a total equilibration time of 22 ns and starting from randomly assigned velocities. The first two equilibration steps were run as NVT ensemble, before switching to the NPT ensemble for the remainder of the equilibration as well as production runs. Temperature coupling at 310 K was achieved using the v-rescale thermostat, while pressure coupling was achieved using the c-rescale barostat." Additionally, some other smaller modifications to this part of the methods section have been made, to improve clarity.

- The MD was used to investigate the distribution of water molecules around the protein. How much this distribution depends on the force field used for the water model?

Response: CHARMM36 is a standard force field used in molecular dynamics (MD) simulations to model the behavior of biological macromolecules, such as proteins, and is particularly well-suited for simulating membrane protein systems. It has been optimized to work with the TIP3P water model, a simple and widely used model in MD simulations, which represents a water molecule with three interaction sites: one for the oxygen and one for each hydrogen atom. While the simplicity of TIP3P can limit the accuracy of hydrogen bonding representation, more sophisticated models like TIP4P and TIP5P, which offer better representations of water's dielectric properties and hydrogen-bonding networks, can negatively affect protein behavior by altering the helix/coil equilibrium. Therefore, to ensure optimal simulation of the protein and observe water occupancy in the internal cavity without speculating on potential water-mediated hydrogen bonds, we adopted a conservative approach using the CHARMM36 force field with the TIP3P water model. (12)

-More references about F GPCRS besides ref. 1 (which is from some of the authors) could be added.

Response: To address this comment, we added multiple references: (13–18)

-There are repetitions in the paper ('The most extensively studied GPCR classes, A, B, and C, contain conserved motifs, allosteric sodium binding sites, and cholesterol binding sites as well as an intricate network of waters that define the dynamic conformational landscape of the receptor. In the case for FZDs, these features remain to be elucidated.' and 'Frizzled ...are class F GPCRs that remain understudied')

Response: To address this comment, we removed the repetition.

REFERENCES

1. E. H. Schneider, R. Seifert, Sf9 cells: A versatile model system to investigate the pharmacological properties of G protein-coupled receptors. *Pharmacol Ther* **128**, 387–418 (2010).
2. S. Saha, B. Khanppnavar, J. Maharana, H. Kim, C. M. C. Carino, C. Daly, S. Houston, P. Kumari, P. N. Yadav, B. Plouffe, A. Inoue, K. Y. Chung, R. Banerjee, V. M. Korkhov, A. K. Shukla, Structure of the human Duffy antigen receptor. *bioRxiv*, 2023.07.09.548245 (2023).
3. A. Xia, X. Yong, C. Zhang, G. Lin, G. Jia, C. Zhao, X. Wang, Y. Hao, Y. Wang, P. Zhou, X. Yang, Y. Deng, C. Wu, Y. Chen, J. Zhu, X. Tang, J. Liu, S. Zhang, J. Zhang, Z. Xu, Q. Hu, J. Zhao, Y. Yue, W. Yan, Z. Su, Y. Wei, R. Zhou, H. Dong, Z. Shao, Y. S, Cryo-EM structures of human GPR34 enable the identification of selective antagonists. *Proc Natl Acad Sci* **120**, 2017 (2023).
4. J. Petersen, S. C. Wright, D. Rodríguez, P. Matricon, N. Lahav, A. Vromen, A. Friedler, J. Strömqvist, S. Wennmalm, J. Carlsson, G. Schulte, Agonist-induced dimer dissociation as a macromolecular step in G protein-coupled receptor signaling. *Nat Commun* **8** (2017).
5. A. Kaykas, J. Yang-Snyder, M. Héroux, K. V. Shah, M. Bouvier, R. T. Moon, Mutant Frizzled 4 associated with vitreoretinopathy traps wild-type Frizzled in the endoplasmic reticulum by oligomerization. *Nat Cell Biol* **6**, 52–58 (2004).
6. M. Kowalski-Jahn, H. Schihada, A. Turku, T. Huber, T. P. Sakmar, G. Schulte, Frizzled BRET sensors based on bioorthogonal labeling of unnatural amino acids reveal WNT-induced dynamics of the cysteine-rich domain. *Sci Adv* **7**, 1–15 (2021).
7. Z. J. DeBruine, J. Ke, K. G. Harikumar, X. Gu, P. Borowsky, B. O. Williams, W. Xu, L. J. Miller, H. E. Xu, K. Melcher, Wnt5a promotes Frizzled-4 signalosome assembly by stabilizing cysteine-rich domain dimerization. *Genes Dev* **31**, 916–926 (2017).
8. A. Turku, H. Schihada, P. Kozielwicz, C. F. Bowin, G. Schulte, Residue 6.43 defines receptor function in class F GPCRs. *Nat Commun* **12**, 1–14 (2021).

9. A. Erazo-Oliveras, M. Muñoz-Vega, M. Mlih, V. Thiriveedi, M. L. Salinas, J. M. Rivera-Rodríguez, E. Kim, R. C. Wright, X. Wang, K. K. Landrock, J. S. Goldsby, D. A. Mullens, J. Roper, J. Karpac, R. S. Chapkin, Mutant APC reshapes Wnt signaling plasma membrane nanodomains by altering cholesterol levels via oncogenic β -catenin. *Nat Commun* **14** (2023).
10. C. F. Bowin, P. Kozielowicz, L. Grätz, M. Kowalski-Jahn, H. Schihada, G. Schulte, WNT stimulation induces dynamic conformational changes in the Frizzled-Dishevelled interaction. *Sci Signal* **16**, 1–18 (2023).
11. D. Hilger, K. K. Kumar, H. Hu, M. F. Pedersen, S. O. Brien, L. Giehm, C. Jennings, G. Eskici, A. Inoue, J. M. Mathiesen, G. Skiniotis, B. K. Kobilka, Structural insights into differences in G protein activation by family A and family B GPCRs. **369**, 1–40 (2021).
12. S. Boonstra, P. R. Onck, E. Van Der Giessen, CHARMM TIP3P Water Model Suppresses Peptide Folding by Solvating the Unfolded State. *Journal of Physical Chemistry B* **120**, 3692–3698 (2016).
13. M. M. Scharf, L. J. Humphrys, S. Berndt, A. Di Pizio, J. Lehmann, I. Liebscher, A. Nicoli, M. Y. Niv, L. Peri, H. Schihada, G. Schulte, The dark sides of the GPCR tree - research progress on understudied GPCRs. *Br J Pharmacol*, 1–26 (2024).
14. G. Schulte, International Union of Basic and Clinical LXXX. The Class Frizzled Receptors. *Pharmacol Rev* **62**, 632–667 (2010).
15. P. Kozielowicz, A. Turku, C. F. Bowin, J. Petersen, J. Valnohova, M. C. A. Cañizal, Y. Ono, A. Inoue, C. Hoffmann, G. Schulte, Structural insight into small molecule action on Frizzleds. *Nat Commun* **11** (2020).
16. G. Schulte, M. M. Scharf, J. Bous, J. H. Voss, L. Grätz, P. Kozielowicz, Frizzleds act as dynamic pharmacological entities. *Trends Pharmacol Sci* **xx**, 1–11 (2024).
17. L. Grätz, M. Kowalski-Jahn, M. M. Scharf, P. Kozielowicz, M. Jahn, J. Bous, N. A. Lambert, D. E. Gloriam, G. Schulte, Pathway selectivity in Frizzleds is achieved by conserved micro-switches defining pathway-determining, active conformations. *Nat Commun* **14**, 4573 (2023).
18. G. Schulte, 75th Anniversary Celebration Collection International Union of Basic and Clinical Pharmacology . CXIV: The Class F of G Protein-Coupled Receptors. *Pharmacol Rev* **76** (2024).

Reviewers' Comments:

Reviewer #1:

Remarks to the Author:

The revised manuscript is improved; All my questions are adequately addressed.

Reviewer #2:

Remarks to the Author:

I thank the authors for addressing all of my comments and for providing an in-depth discussion about the role of cholesterol in FZD receptor biology. I congratulate them on this excellent work!

Reviewer #3:

Remarks to the Author:

The authors properly addressed all my issues and they modified the text accordingly for all of these issues except one:

"CHARMM36 is a standard force field used in molecular dynamics (MD) simulations to model the behavior of biological macromolecules, such as proteins, and is particularly well-suited for simulating membrane protein systems. It has been optimized to work with the TIP3P water model, a simple and widely used model in MD simulations, which represents a water molecule with three interaction sites: one for the oxygen and one for each hydrogen atom. While the simplicity of TIP3P can limit the accuracy of hydrogen bonding representation, more sophisticated models like TIP4P and TIP5P, which offer better representations of water's dielectric properties and hydrogen-bonding networks, can negatively affect protein behavior by altering the helix/coil equilibrium. Therefore, to ensure optimal simulation of the protein and observe water occupancy in the internal cavity without speculating on potential water-mediated hydrogen bonds, we adopted a conservative approach using the CHARMM36 force field with the TIP3P water model. (12)"

The authors should include also this point in their manuscript.

Rebuttal letter

In the text below we address the reviewer's comments, which were very helpful to guide the revision efforts. Our responses are shown in **green**.

Remaining reviewers' comments:

Reviewer #1

The revised manuscript is improved; All my questions are adequately addressed.

We are glad that we could address all the criticism raised.

Reviewer #2

I thank the authors for addressing all of my comments and for providing an in-depth discussion about the role of cholesterol in FZD receptor biology. I congratulate them on this excellent work!

Thanks a lot for the positive comments and the constructive review overall!

Reviewer #3

The authors properly addressed all my issues and they modified the text accordingly for all of these issues except one:

"CHARMM36 is a standard force field used in molecular dynamics (MD) simulations to model the behavior of biological macromolecules, such as proteins, and is particularly well-suited for simulating membrane protein systems. It has been optimized to work with the TIP3P water model, a simple and widely used model in MD simulations, which represents a water molecule with three interaction sites: one for the oxygen and one for each hydrogen atom. While the simplicity of TIP3P can limit the accuracy of hydrogen bonding representation, more sophisticated models like TIP4P and TIP5P, which offer better representations of water's dielectric properties and hydrogen-bonding networks, can negatively affect protein behavior by altering the helix/coil equilibrium. Therefore, to ensure optimal simulation of the protein and observe water occupancy in the internal cavity without speculating on potential water-mediated hydrogen bonds, we adopted a conservative approach using the CHARMM36 force field with the TIP3P water model. (12)"

The authors should include also this point in their manuscript.

We have now implemented this information in the manuscript. Thanks for the recommendation.